# Valence biases in reinforcement learning shift across adolescence and modulate subsequent memory

**Gail M Rosenbaum[1], Hannah L Grassie[1], Catherine A Hartley[1,2]***

[1]Department of Psychology, New York University, New York, United States; [2]Center for Neural Science, New York University, New York, United States

**Abstract** As individuals learn through trial and error, some are more influenced by good outcomes, while others weight bad outcomes more heavily. Such valence biases may also influence memory for past experiences. Here, we examined whether valence asymmetries in reinforcement learning change across adolescence, and whether individual learning asymmetries bias the content of subsequent memory. Participants ages 8–27 learned the values of 'point machines,' after which their memory for trial-unique images presented with choice outcomes was assessed. Relative to children and adults, adolescents overweighted worse-than-expected outcomes during learning. Individuals' valence biases modulated incidental memory, such that those who prioritized worse- (or better-) than-expected outcomes during learning were also more likely to remember images paired with these outcomes, an effect reproduced in an independent dataset. Collectively, these results highlight age-related changes in the computation of subjective value and demonstrate that a valence-asymmetric valuation process influences how information is prioritized in episodic memory.

## Editor's evaluation

This paper will be of interest to cognitive and behavioral neuroscientists, behavioral economists, and developmental psychologists. The authors provide novel evidence that adolescents, relative to children and young adults, are prone to making risk-averse decisions because they are more attuned to negative outcomes during learning. The paper presents rigorous computational analyses that conclusively support the major claims and advance our understanding of age-related shifts in decision making.

**\*For correspondence:**
cah369@nyu.edu

**Competing interest:** The authors declare that no competing interests exist.

## Introduction

Throughout our lives, we encounter many new or uncertain situations in which we must learn, through trial and error, which actions are beneficial and which are best avoided. Determining which behaviors will earn praise from a teacher, which social media posts will be liked by peers, or which route to work will have the least traffic is often accomplished by exploring different actions, and learning from the good or bad outcomes that they yield. Importantly, individuals differ in the extent to which their evaluations (*Daw et al., 2002*; *Frank et al., 2004*; *Gershman, 2015*; *Lefebvre et al., 2017*; *Sharot and Garrett, 2016*) and their memories (*Madan et al., 2014*, *Madan et al., 2017*; *Rouhani and Niv, 2019*) are influenced by good versus bad experiences. For example, consider a diner who has a delicious meal on her first visit to a new sushi restaurant, but on her next visit, the meal is not very good. A tendency to place greater weight on past positive experiences might make her both more likely to remember the good dining experience and more likely to return and try the restaurant again. In contrast, if the recent negative experience exerts an outsized influence, it may be more easily called

to mind and she may forego another visit to that restaurant in favor of a surer bet. In this manner, asymmetric prioritization of past positive versus negative outcomes may render these valenced experiences more persistent in our memories and systematically alter how we make future decisions about uncertain prospects.

Understanding how experiential learning informs decision-making under uncertainty may be particularly important during adolescence, when teens' burgeoning independence offers more frequent exposure to novel contexts in which the potential positive or negative outcomes of an action may be uncertain. Epidemiological data reveal an adolescent peak in the prevalence of many 'risky' behaviors that carry potential negative consequences (e.g., criminal behavior [*Steinberg, 2013*], risky sexual behavior [*Satterwhite et al., 2013*]). Moreover, consistent with proposals that adolescent risk taking might be driven by heightened sensitivity to rewarding outcomes (*Casey et al., 2008*; *Galván, 2013*; *Silverman et al., 2015*; *Steinberg, 2008*; *van Duijvenvoorde et al., 2017*), several neuroimaging studies have observed that adolescents exhibit neural responses to reward that are greater in magnitude than those of children or adults (*Braams et al., 2015*; *Cohen et al., 2010*; *Galvan et al., 2006*; *Silverman et al., 2015*; *Van Leijenhorst et al., 2010*). These findings suggest that as adolescents learn to evaluate novel situations through trial and error, positive experiences might exert an outsized influence on their subsequent actions and choices.

Reinforcement learning (RL) models mathematically formalize the process of evaluating actions based on their resulting good and bad outcomes (*Sutton and Barto, 1998*). In such models, action value estimates are iteratively revised based on prediction errors or the extent to which an experienced outcome deviates from one's current expectation. The magnitude of the resulting value update is scaled by an individual's learning rate. Valence asymmetries in the estimation of action values can be captured by positing two distinct learning rates for positive versus negative prediction errors, leading to differential adjustment of value estimates following outcomes that are better or worse than one's expectations. Importantly, an RL algorithm with such valence-dependent learning rates estimates subjective values in a 'risk-sensitive' manner (*Mihatsch and Neuneier, 2002*; *Niv et al., 2012*). A learner with a greater positive than negative learning rate will, across repeated choices, come to assign a greater value to a risky prospect (i.e., with variable outcomes) than to a safer choice with equivalent expected value (EV) that consistently yields intermediate outcomes, whereas a learner with the opposite asymmetry will estimate the risky option as being relatively less valuable.

Outcomes that violate our expectations might also be particularly valuable to remember. Beyond the central role of prediction errors in the estimation of action values, these learning signals also appear to influence what information is prioritized in episodic memory (*Ergo et al., 2020*). Past studies have demonstrated enhanced memory for stimuli presented concurrently with outcomes that elicit positive (*Davidow et al., 2016*; *Jang et al., 2019*), negative (*Kalbe and Schwabe, 2020*), or high-magnitude (independent of valence) prediction errors (*Rouhani et al., 2018*), suggesting that prediction errors can facilitate memory encoding and consolidation processes. The common role of prediction errors in driving value-based learning and facilitating memory may reflect, in part, a tendency to allocate greater attention to stimuli that are uncertain (*Dayan et al., 2000*; *Pearce and Hall, 1980*). However, it is unclear whether idiosyncratic valence asymmetries in RL computations might give rise to corresponding asymmetries in the information that is prioritized for memory. Moreover, while few studies have explored the development of these interactive learning systems, a recent empirical study observing an effect of prediction errors on recognition memory in adolescents, but not adults (*Davidow et al., 2016*), suggests that the influence of RL signals on memory may be differentially tuned across development.

In the present study, we examined whether valence asymmetries in RL change across adolescent development, conferring age differences in risk preferences. We additionally hypothesized that individuals' learning asymmetries might asymmetrically bias their memory for images that coincide with positive versus negative prediction errors. Several past studies have characterized developmental changes in learning from valenced outcomes (*Christakou et al., 2013*; *Hauser et al., 2015*; *Jones et al., 2014*; *Master et al., 2020*; *Moutoussis et al., 2018*; *van den Bos et al., 2012*). However, the probabilistic reinforcement structures used in each of these studies demanded that the learner adopt specific valence asymmetries during value estimation in order to maximize reward in the task (*Nussenbaum and Hartley, 2019*). For instance, in one study, child, adolescent, and adult participants were rewarded on 80% of choices for one option and 20% of choices for a second option (*van den*

*Bos et al., 2012*). In this task, a positive learning asymmetry yields better performance than a neutral or negative asymmetry (*Nussenbaum and Hartley, 2019*). Indeed, adults exhibited a more optimal pattern of learning, with higher positive than negative learning rates, while children and adolescents did not (*van den Bos et al., 2012*). Thus, choice behavior in these studies might reflect both potential age differences in the optimality of RL, as well as context-independent differences in the weighting of positive versus negative prediction errors (*Cazé and van der Meer, 2013*; *Nussenbaum and Hartley, 2019*).

In Experiment 1 of the present study, we assessed whether valence asymmetries in RL varied from childhood to adulthood, using a risk-sensitive RL task (*Niv et al., 2012*) in which probabilistic and deterministic choice options have equal EV, making no particular learning asymmetry optimal. This parameterization allows any biases in the weighting of positive versus negative prediction errors to be revealed through subjects' systematic risk-averse or risk-seeking choice behavior. Each choice outcome in the task was associated with a trial-unique image, enabling assessment of whether valenced learning asymmetries also biased subsequent memory for images that coincided with good or bad outcomes.

To determine whether this hypothesized correspondence between valence biases in learning and memory generalized across experimental tasks and samples of different ages, in Experiment 2, we conducted a reanalysis of data from a previous study (*Rouhani et al., 2018*). In this study, a group of adults completed a task in which they reported value estimates for a series of images, and later completed a memory test for the images they encountered during learning. The original manuscript reported that subsequent memory varied as a function of PE magnitude, but not valence. Here, we tested whether a valence-dependent effect of PE on memory might be evident after accounting for idiosyncratic valence biases in learning.

## Results

### Experiment 1

Participants (N = 62) ages 8–27 (M = 17.63, SD = 5.76) completed a risk-sensitive RL task (*Niv et al., 2012*). In this task, participants learned, through trial and error, the values and probabilities associated with probabilistic and deterministic 'point machines' (*Figure 1A and B*). On each trial (183 trials), participants made a free (two-choice options) or forced (single-choice option) selection of a

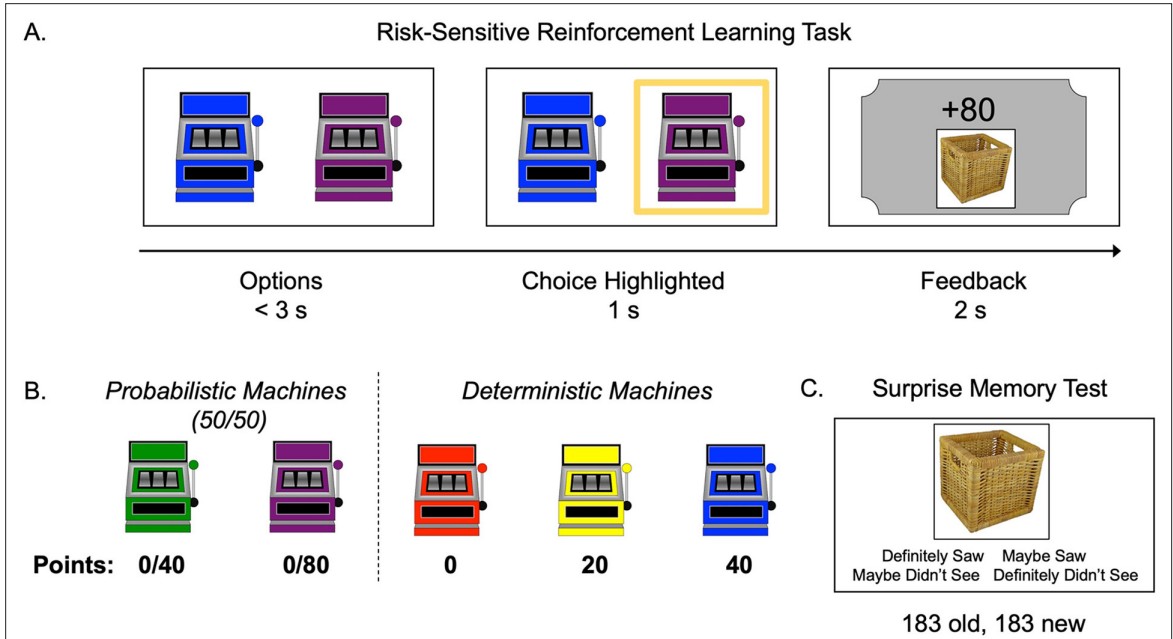

**Figure 1.** Task structure. (**A**) Schematic of the structure of a trial in the risk-sensitive reinforcement learning task. (**B**) The probabilities and point values associated with each of five 'point machines' (colors were counterbalanced). (**C**) Example memory trial.

point machine. Within free-choice trials, 'risky' trials presented a pair consisting of one probabilistic and one deterministic option, where neither option strictly dominated the other and evidence of individuals' subjective values was revealed by their choices. On 'test' trials, in which one option dominated the other, we could assess objectively the accuracy of participants' learning. We presented feedback (number of points) from each choice on a 'ticket' that also displayed a trial-unique picture of an object. A subsequent memory test allowed us to explore the interaction between choice outcomes and memory encoding across age (*Figure 1C*).

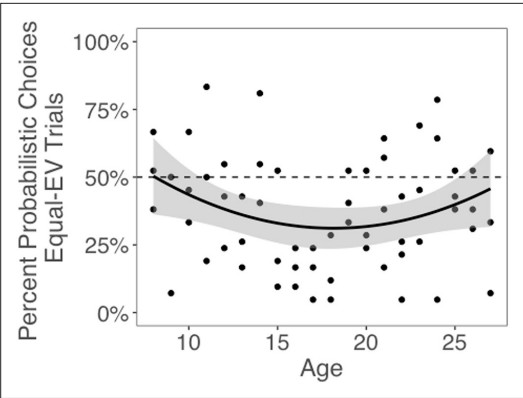

**Figure 2.** Probabilistic choices by age. Probabilistic (i.e., risky) choices by age on trials in which the risky and safe machines had equal expected value (EV). Data points depict the mean percentage of trials where each participant selected the probabilistic choice option as a function of age. The regression line is from a linear regression including linear and quadratic age terms (significant quadratic effect of age: $b = 0.06$, 95% CI [0, 0.12], $t(59) = 2.14$, p=0.036, $f^2 = .08$, 95% CI [0, 0.29], N = 62). Shaded region represents 95% CIs for estimates.

## Test trial performance

To ensure that participants learned the probabilities and outcomes associated with each machine, we first examined performance on test trials, in which one option dominated the other. Test trial accuracy significantly improved across the task (generalized linear mixed-effects model: $z = 8.56$, p<0.001, OR = 2.03, 95% CI [1.72, 2.38]), with accuracy improving from a mean of 0.63 in the first block to means of 0.80 and 0.84 in blocks 2 and 3, respectively. There was no main effect of age ($z = 0.51$, p=0.612, OR = 1.06, 95% CI [0.86, 1.30]) or interaction between age and trial number ($z = 0.22$, p=0.830, OR = 1.02, 95% CI [0.87, 1.19]; *Appendix 1—figure 1A*). These results suggest that accuracy on this coarse measure of value learning did not change with age in our task.

## Explicit reports

Following the learning task, we probed participants' explicit knowledge about the point machines. Consistent with participants' high accuracy on test trials, accuracy was also high on participants' reports of whether each point machine was probabilistic or deterministic (*M* = 0.85) and for the point values associated with each machine (*M* = 0.84). Linear regressions suggested that performance on these explicit accuracy metrics did not vary with linear age (probabilistic/deterministic response accuracy by age: $b = -0.02$, 95% CI [−0.06, 0.03], $t(60) = -0.88$, p=0.382, $f^2 = 0.01$, 95% CI [0, 0.13]; point value response accuracy by age: $b = 0.02$, 95% CI [−0.04, 0.07], $t(60) = 0.65$, p=0.516, $f^2 = 0.01$, 95% CI [0, 0.11]).

## Response time

We explored whether response time (RT) varied with age during the learning task. We found a significant interaction between age and trial number (linear mixed-effects model: $t(11279) = -2.10$, p=0.036, $b = -0.02$, 95% CI [−0.04, 0]) predicting log-transformed RT. Although RT did not differ by age early in the experiment, older participants responded faster than younger participants by the end of the experiment.

## Decision-making

Importantly, in our task, there were two pairs of machines in which both probabilistic and deterministic options yielded the same EV (i.e., 100% 20 points and 50/50 0/40 points; 100% 40 points and 50/50 0/80 points). A primary goal of this study was to examine participants' tendency to choose probabilistic versus deterministic machines when EV was equivalent. On these equal-EV risk trials, participants chose the probabilistic option on 37% of trials (SD = 21%). This value was significantly lower than 50% (one-sample *t*-test: $t(61) = 4.87$, p<0.001, $d = 0.62$, 95% CI [0.37, 0.95]), suggesting

that, despite exhibiting heterogeneity in risk preferences, participants as a group were generally risk averse.

Next, we tested whether choices of the probabilistic machines, compared to choices for equal-EV deterministic machines, changed with age. The best-fitting model included both linear and quadratic age terms ($F(1,59)$ = 4.58, p=0.036), indicating that risk taking changed nonlinearly with age. Contrary to our hypothesis that risk-seeking choices would be highest in adolescents, we observed a significant quadratic effect of age, such that adolescents chose the probabilistic options less often than children or adults (quadratic age effect in a linear regression including both linear and quadratic age terms: $b$ = 0.06, 95% CI [0, 0.12], $t(59)$ = 2.14, p=0.036, $f^2$ = 0.08, 95% CI [0, 0.29]; *Figure 2*; see *Appendix 1—figure 1* for plots depicting risk taking across the task as a function of age). The linear effect of age was not significant ($b$ = –0.01, 95% CI [–0.06, 0.12], $t(59)$ = –0.44, p=0.662, $f^2$ =

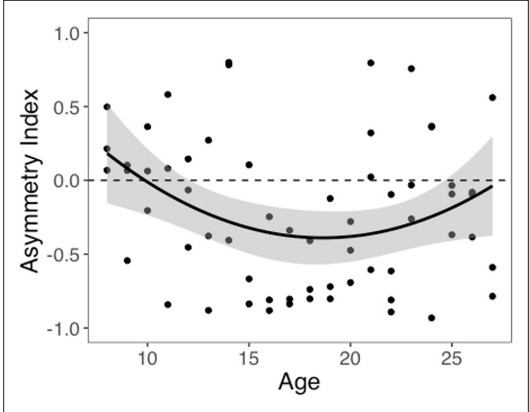

**Figure 3.** Asymmetry index (AI) by age. The regression line is from a linear regression model including linear and quadratic age terms ($b$ = 0.17, 95% CI [0.03, 0.31], $t(59)$ = 2.43, p=0.018, $f^2$ = 0.10, 95% CI [0, 0.33], N = 62). Data points represent individual participants. Shaded region represents 95% CIs for estimates.

0.01, 95% CI [0, 0.11]). We also conducted a regression using the two-lines approach (*Simonsohn, 2018*) and found a significant u-shaped pattern of risk taking with age, where the proportion of probabilistic choices decreased from age 8–16.45 ($b$ = –0.03, $z$ = –1.97, p=0.048) and increased from age 16.45–27 ($b$ = 0.02, $z$ = 1.98, p=0.048). Age patterns were qualitatively similar when considering the subset of trials in which participants faced choice options with unequal EV (i.e., the 0/80 point machine vs. the safe 20 point machine; see Appendix 1 and *Appendix 1—figure 2* for full results).

## Reinforcement learning modeling

To better understand the learning processes underlying individuals' decision-making, we compared the fit of four RL models to participants' choice behavior. The first was a temporal difference (TD) model with one learning rate ($\alpha$). The second was a risk-sensitive temporal difference (RSTD) model with separate learning rates for better-than-expected ($\alpha^+$) and worse-than-expected ($\alpha^-$) outcomes, allowing us to index valence biases in learning. The third model included four learning rates (FourLR), with separate $\alpha^+$ and $\alpha^-$ for free and forced choices, as past studies have found learning may differ as a function of agency (*Chambon et al., 2020*; *Cockburn et al., 2014*). Finally, the fourth model was a Utility model, which transforms outcome values into utilities with an exponential subjective utility function with a free parameter ($\rho$) capturing individual risk preferences (*Pratt, 1964*), updated value estimates using a single learning rate. For all models, machine values were transformed to range from 0 to 1, and values were initialized at 0.5 (equivalent to 40 points). A softmax function with an additional parameter $\beta$ was used to convert the relative estimated values of the two machines into a probability of choosing each machine presented for maximum likelihood estimation.

The RSTD (median Bayesian information criterion (BIC) = 131.93) and Utility (median BIC = 131.06) models both provided a better fit to participants' choice data than both the TD (median BIC = 145.35) and FourLR (median BIC = 141.25) models (*Appendix 1—figure 5*). Assessment of whether the RSTD or Utility model provided the best fit to participants' data was equivocal. At the group level, median ΔBIC was 0.87, while at the subject level, the median ΔBIC was 0.33. Thus, neither ΔBIC metric provides clear evidence in favor of either model (ΔBIC > 6 ; *Raftery, 1995*).

To further arbitrate between the RSTD and Utility models, we ran posterior predictive checks and confirmed that simulations from both models generated using subjects' fit parameter values yielded choice behavior that exhibited strong correspondence to the real participant data (see *Appendix 1—figure 9*). However, data simulated from the RSTD model exhibited a significantly stronger correlation with actual choices ($r$ = 0.92) than those simulated using the Utility model ($r$ = 0.89; $t(61)$ = 2.58, p=0.012). Because the RSTD model fit choice data approximately as well as the Utility model,

provided a significantly better qualitative fit to the choice data, and yielded an index of valence biases in learning, we focused our remaining analyses on the RSTD model (see Appendix 1 for additional model comparison analyses and for an examination of the relation between the Utility model and subsequent memory data).

We computed an asymmetry index (AI) for each participant, which reflects the relative size of $\alpha^+$ and $\alpha^-$, from the RSTD model. Mean AI was –0.22 (SD = 0.50). Mirroring the age patterns observed in risk taking, a linear regression model with a quadratic age term fit better than the model with only linear age ($F$(1,59) = 5.88, p=0.018), and there was a significant quadratic age pattern in AI ($b$ = 0.17, 95% CI [0.03, 0.31], $t$(59) = 2.43, p=0.018, $f^2$ = 0.10, 95% CI [0, 0.33]; *Figure 3*). Further, the u-shaped relationship between AI and age was significant, with a decrease in AI from ages 8–17 ($b$ = –0.08, $z$ = –3.82, p<0.001), and an increase from ages 17–27 ($b$ = 0.05, $z$ = 2.17, p=0.030). This pattern was driven primarily by age-related changes in $\alpha^-$, which was greater in adolescents relative to children and adults (better fit for linear regression including quadratic term: $F$(1,59) = 9.04, p=0.004; quadratic age: $b$ = –0.09, 95% CI [–0.16, –0.03], $t$(59) = –3.01, p=0.004, $f^2$ = 0.15, 95% CI [0.02, 0.43]; *Appendix 1—figure 10B*). According to the two-lines approach, $\alpha^-$ significantly increased from ages 8–18 ($b$ = 0.04, $z$ = 3.24, p=0.001) and decreased from ages 18–27 ($b$ = –0.04, $z$ = –3.57, p<0.001). Conversely, there were no linear or quadratic effects of age for $\alpha^+$ (all $p$s>0.24; *Appendix 1—figure 10A*). Finally, there were no significant linear or quadratic age patterns in the $\beta$ parameter ($p$s>.15, see Appendix 1 for full results; *Appendix 1—figure 10C*).

Prior work has found that valence biases tend to be positive in free choices, but neutral or negative in forced choices (*Chambon et al., 2020*; *Cockburn et al., 2014*). While model comparison indicated that the FourLR model did not provide the best account of participants' learning process, we nonetheless conducted an exploratory analysis in which we used parameter estimates from the FourLR model to test whether learning asymmetries varied as a function of agency in our study. While the $\alpha+$ and AI were both higher for free compared to forced trials, median AIs were negative for both free and forced choices (see Appendix 1 for full results; *Appendix 1—figure 12*).

## Memory performance

Next, we examined accuracy during the surprise memory test for images that were presented with choice outcomes. Participants correctly identified 54% (SD = 14%) of images presented alongside choice feedback (i.e., Hits) and incorrectly indicated that 24% (SD = 15%) of foil images had been presented during the choice task (False Alarms). Mean $d'$ was 0.93 (SD = 0.48). Hit rate did not significantly change with linear or quadratic age ($p$s>0.14). However, false alarm rate significantly increased with linear age (linear regression: $b$ = 0.04, 95% CI [0.00; 0.08], $t$(60) = 2.14, p=0.037, $f^2$ = 0.08, 95% CI [0, 0.28]; *Appendix 1—figure 3A*). There was a marginal linear decrease in $d'$ with age (linear regression: $b$ = –0.11, 95% CI [–0.23, 0.01], $t$(60) = 1.84, p=0.070, $f^2$ = 0.06, 95% CI [0, 0.24]; *Appendix 1—figure 3B*), suggesting that adults performed slightly worse on the memory test than younger participants.

## Influence of choice context on memory

We next tested whether the decision context in which images were presented influenced memory encoding. To explore this possibility, we first tested whether participants preferentially remembered images presented with outcomes of probabilistic versus deterministic machines. Participants were significantly more likely to remember pictures presented following a choice that yielded probabilistic rather than deterministic outcomes (probabilistic: $M$ = 0.56, SD = 0.15; deterministic: $M$ = 0.52, $SD$ = 0.15; $t$(61) = 3.08, p=0.003, $d$ = 0.39, 95% CI [0.13, 0.65]). This result suggests that pictures were better remembered when they followed the choice of a machine that consistently generated prediction errors (PEs), which may reflect preferential allocation of attention toward outcomes of uncertain choices (*Dayan et al., 2000*; *Pearce and Hall, 1980*).

Next, we explored whether valence biases in learning could account for individual variability in subsequent memory. In theory, larger-magnitude PEs provide stronger learning signals. Thus, we hypothesized that participants would have better memory for items coinciding with larger PEs. We also expected that this effect might differ as a function of idiosyncratic valence biases, with participants preferentially remembering items coinciding with signed PEs where the sign was consistent with the valence bias of their AI. Of note, this model did not explicitly include a variable indicating

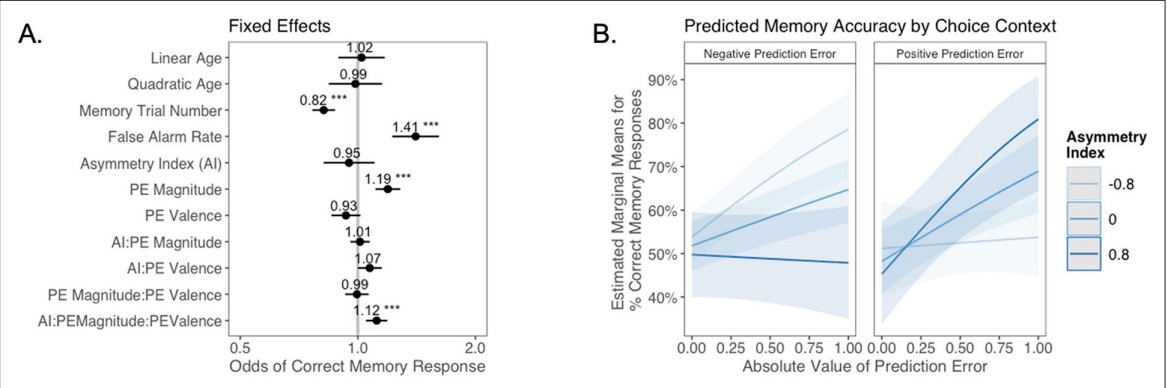

**Figure 4.** The relation between valence biases in learning and incidental memory for pictures presented with choice outcomes (Experiment 1). (**A**) Results from generalized mixed-effects regression depicting fixed effects on memory accuracy. Whiskers represent 95% CI. (**B**) Estimated marginal means plot showing the three-way interaction between AI, PE valence, and PE magnitude (z = 3.45, p=0.001, OR = 1.12, 95% CI [1.05, 1.19], N = 62). Individuals with higher AIs were more likely to remember images associated with larger positive PEs, and those with lower AIs were more likely to remember images associated with larger negative PEs. Shaded areas represent 95% CI for estimates. ***p < .001.

whether outcomes followed probabilistic or deterministic choices. Rather, whether the choice was probabilistic or deterministic was reflected in the PE magnitude variable, which was typically higher for probabilistic choices. In a generalized linear mixed-effects model, we predicted memory accuracy as a function of AI, PE valence, PE magnitude, and their interaction. We also tested for effects of linear and quadratic age, false alarm rate, as a measure of participants' tendency to generally deem items as old, and trial number in the memory task, to account for fatigue as the task progressed (*Figure 4A*). We had no a priori hypothesis about how any effect of valence bias on memory might interact with participants' confidence in their 'old' and 'new' judgments. Therefore, consistent with prior research examining memory accuracy (e.g., *Dunsmoor et al., 2015*; *Murty et al., 2016*), we collapsed across 'definitely' and 'maybe' confidence ratings for our primary analysis (but see Appendix 1 for an exploratory ordinal regression analysis).

As expected, accuracy was significantly higher for those with a higher false alarm rate (suggesting a bias towards old responses; z = 4.86, p<0.001, OR = 1.41, 95% CI [1.23, 1.61]), and accuracy decreased as the task progressed (main effect of memory trial number: z = –5.83, p<0.001, OR = 0.82, 95% CI [0.76, 0.87]). There was a significant effect of unsigned PE magnitude on memory (z = 4.75, p<0.001, OR = 1.19, 95% CI [1.11, 1.28]), such that images that coincided with larger PEs were better remembered. There was also a significant three-way interaction between PE magnitude, PE valence, and AI on memory accuracy (z = 3.45, p=0.001, OR = 1.12, 95% CI [1.05, 1.19]), such that people with more positive AIs were more likely to remember images associated with larger positive PEs (*Figure 4B*). The converse was also true: those with lower AIs were more likely to remember images presented concurrently with outcomes that elicited higher-magnitude negative PEs. Ordinal regression results that considered all four levels of confidence in recollection judgments (*Supplementary file 1*, *Appendix 1—figure 4*) yielded consistent results and suggested that effects were primarily driven by high-confidence responses. Notably, neither linear (z = 0.32, p=0.750, OR = 1.02, 95% CI [0.89, 1.17])

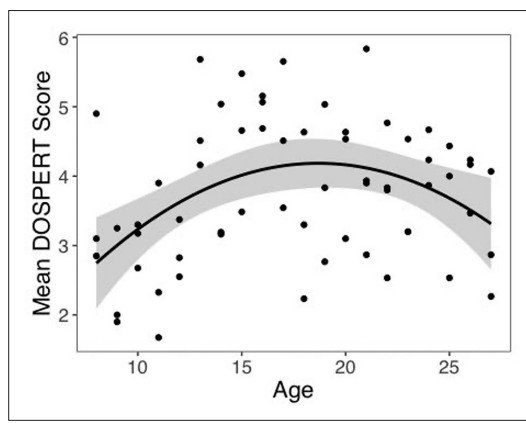

**Figure 5.** Self-reported risk taking by age. Self-reported risk taking on the Domain-Specific Risk Taking (DOSPERT) scale changed nonlinearly with age (linear regression: b = –0.42, 95% CI [–0.69,–0.15], t(59) = –3.09, p=0.003, f² = 0.16, 95% CI [0.02, 0.44], N = 62). Shaded region represents 95% CIs for estimates.

nor quadratic age ($z$ = –0.18, p=0.856, OR = 0.99, 95% CI [0.84, 1.15]) were significant predictors of memory, suggesting that AI parsimoniously accounted for individual differences in memory.

To test whether differences in memory for outcomes of deterministic versus probabilistic trials might have driven the observed AI × PE magnitude × PE valence interaction effect, we reran the regression model only within the subset of trials in which participants made probabilistic choices. Our results did not change — we observed both a main effect of PE magnitude ($z$ = 2.22, p=0.026, OR = 1.11, 95% CI [1.01, 1.23], N = 62) and a significant PE valence × PE magnitude × AI interaction ($z$ = 2.34, p=0.019, OR = 1.11, 95% CI [1.02, 1.21], N = 62).

Finally, we tested for effects of agency — whether an image coincided with the outcome of a free or forced choice — on memory performance. We did not find a significant main effect of agency on memory, and agency did not significantly modulate the AI × PE magnitude × PE valence interaction effect (see Appendix 1 for full results; *Appendix 1—figure 13*).

## Self-reported risk taking

One possible explanation for our unexpected u-shaped relationship between age and risk preferences in our choice task is that the adolescents in our sample might have been atypically risk averse. To investigate this possibility, we examined the relation between age and self-reported risk taking to the Domain-Specific Risk Taking (DOSPERT) scale (*Blais and Weber, 2006*). A linear regression model including quadratic age was a better fit than the model including linear age alone ($F$(1,59) = 9.55, p=0.003). Specifically, consistent with prior reports of increased self-reported risk taking in adolescents, we found a significant inverted u-shaped quadratic age pattern (*Figure 5*, $b$ = –0.42, 95% CI [-0.69, –0.15], $t$(59) = –3.09, p=0.003, $f^2$ = 0.16, 95% CI [0.02, 0.44]). There was not a significant linear age pattern in self-reported risk taking ($b$ = 0.15, 95% CI [–0.09, 0.39], $t$(59) = 1.27, p=0.208, $f^2$ = 0.04, 95% CI [0, 0.20]). A two-lines regression analysis indicated that risk taking increased until age 15.29 ($b$ = 0.23, $z$ = 2.20, p=0.028) and decreased thereafter ($b$ = –0.09, $z$ = –2.03, p=0.042). Despite the fact that both choices in our task and self-report risk taking exhibited nonlinear age-related changes, there was not a significant correlation between DOSPERT score and risk taking in the task ($r$ = –0.12, 95% CI [–0.36, 0.13], $t$(60) = –0.95, p=0.347).

## Experiment 2

Next, we assessed the generalizability of the observed effect of valence biases in learning on memory by conducting a reanalysis of a previously published independent dataset from a study that used a different experimental task in an adult sample (*Rouhani et al., 2018*). Notably, results from this study suggested that unsigned PEs (i.e., PEs of greater magnitude, whether negative or positive) facilitated subsequent memory, but no signed effect was observed. Here, we examined whether signed

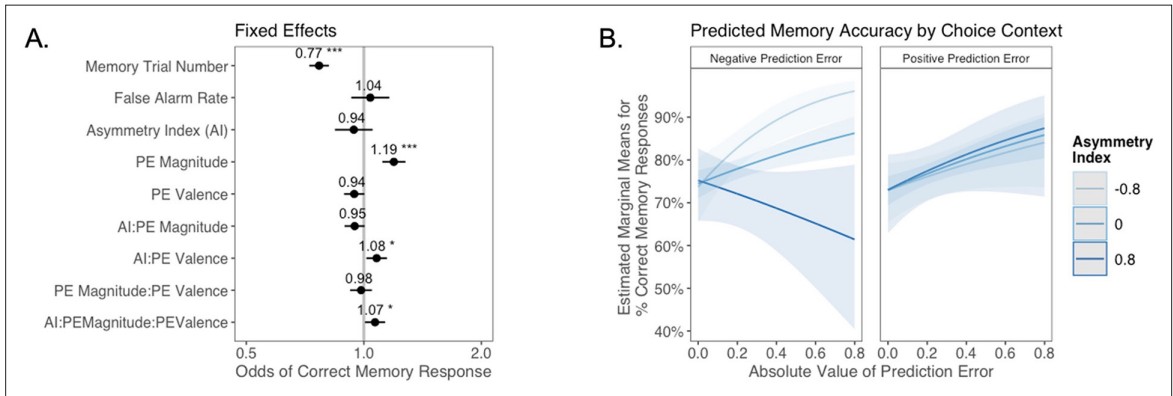

**Figure 6.** The relation between valence biases in learning and incidental memory for pictures presented with trial outcomes (Experiment 2). Reanalysis of data from *Rouhani et al., 2018*. (**A**) Results from generalized mixed-effects regression depicting fixed effects on memory accuracy. Whiskers represent 95% CI. (**B**) Estimated marginal means plot showing the three-way interaction between AI, PE valence, and PE magnitude ($z$ = 2.19, p=0.029, OR = 1.07, 95% CI [1.01, 1.13], N = 305). Individuals with higher AIs were more likely to remember images associated with larger positive PEs, and those with lower AIs were more likely to remember images associated with larger negative PEs. Shaded areas represent 95% CI for estimates. *p < .05, ***p < .001.

valence-specific effects might be evident when we account for individual differences in valence biases in learning.

Participants (N = 305) completed a Pavlovian learning task in which they encountered indoor and outdoor scenes. One type of scene had higher average value than the other. On each trial, an indoor or outdoor image was displayed, and participants provided an explicit prediction for the average value of that scene type. After the learning task, participants completed a memory test for the scenes.

To quantify valence biases in this task, we fit an 'Explicit Prediction' RL model that was similar to the RSTD model used in Experiment 1, but was fit to participants' trial-by-trial predictions rather than to choices. Like RSTD, the Explicit Prediction model included $\alpha^+$ and $\alpha^-$, allowing us to quantify each participant's AI based on the relative size of their best-fit $\alpha^+$ and $\alpha^-$ parameters. Mean AI was –0.11 (SD = 0.34). Next, we ran a generalized linear mixed-effects model, as in Experiment 1, to examine whether PE valence and magnitude interacted with AI to predict subsequent memory, controlling for false alarm rate and memory trial number. Results are reported in *Figure 6*.

Consistent with the results reported in the original manuscript (*Rouhani et al., 2018*), as well as the findings in Experiment 1, there was a strong main effect of unsigned PE (i.e., PE magnitude) on memory (z = 5.09, p<0.001, OR = 1.19, 95% CI [1.12, 1.28]). However, aligned with our results from Experiment 1, we also observed a significant three-way interaction between AI, PE magnitude, and PE valence (z = 2.19, p=0.029, OR = 1.07, 95% CI [1.01, 1.13]). Qualitative examination of this interaction effect suggests that the pattern of results differed slightly from that in Experiment 1. In Experiment 2, differences in memory performance as a function of AI were primarily apparent for images coinciding with negative PEs (*Figure 6B*): those who learned more from negative PEs also had better episodic memory for images that coincided with increasingly large negative PEs, while all participants appeared to have stronger memory for images coinciding with larger positive PEs. Notably, the interaction pattern here mirrors that within the subset of forced trials from Experiment 1 (*Appendix 1— figure 13B*) where, as in Experiment 2, participants learned from observed outcomes, but did not make free choices. One possibility is that PE magnitude and PE valence enhance memory through separate mechanisms, with a universal positive effect of unsigned PEs but a contextually (depending on choice agency) and individually variable effect of PE valence.

## Discussion

In this study, we examined whether asymmetry in learning from good versus bad choice outcomes changed across adolescence, and whether valence biases in RL also influenced episodic memory encoding. Specifically, we hypothesized that adolescents would place greater weight on good than bad outcomes during learning, a potential cognitive bias that may contribute to the increased risk taking during adolescence evident in real-world epidemiological data (*Kann et al., 2018*). We indeed observed nonlinear age differences in valence-based learning asymmetries, but in the direction opposite from our prediction. Adolescents learned more from outcomes that were worse than expected, which was reflected in less risk taking relative to children and adults. Within this developmental sample, individual differences in learning biases were mirrored in subsequent memory. People who learned more from surprising negative versus positive outcomes also had better memory for images that coincided with negative outcomes, and vice versa. Although the precise pattern of results differed slightly, this relation between idiosyncratic valence biases in RL and corresponding biases in subsequent memory was also evident in a second independent sample in a different task (*Rouhani et al., 2018*), suggesting that this finding is generalizable. Collectively, these results highlight age-related changes across adolescence in the computation of subjective value and demonstrate that an individually varying valence asymmetric valuation process also influences how information is prioritized in memory.

Age-related shifts in learning rate asymmetry were driven primarily by changes in negative, rather than positive, learning rates. Whereas negative learning rates changed nonlinearly with age, there was no evidence for significant age differences in positive learning rates. This absence of age-related change in reward learning may seem counterintuitive given a large literature characterizing heightened reward sensitivity in adolescence (for reviews, see *Galván, 2013*; *Silverman et al., 2015*; *van Duijvenvoorde et al., 2016*); however, these effects have largely been observed in tasks in which learning was not required. Moreover, heightened reactivity to negatively valenced stimuli has also been observed in adolescents, relative to children (*Master et al., 2020*) and adults (*Galván and*

*McGlennen, 2013*), and adolescents have been found to exhibit greater sensitivity to negative social evaluative feedback than adults (*Rodman et al., 2017*). While a relatively small number of studies have used RL models to characterize age-related changes in valence-specific value updating (*Christakou et al., 2013*; *Hauser et al., 2015*; *Jones et al., 2014*; *Master et al., 2020*; *Moutoussis et al., 2018*; *van den Bos et al., 2012*), age patterns reported in these studies vary substantially and none observed the same pattern of valence asymmetries present in our data. Variability in these findings may be due in part to substantial variation in the task reward structures, each of which required specific asymmetric settings of learning rates in order to perform optimally (*Cazé and van der Meer, 2013*; *Nussenbaum and Hartley, 2019*). This task variation limits the ability to differentiate age differences in optimal learning from systematic age differences in the influence of positive versus negative prediction errors on subjective value computation (*Nussenbaum and Hartley, 2019*). In contrast, our study used a paradigm in which risky and safe options had equal EV, allowing us to index risk preferences and corresponding valence biases in a context where there was no optimal strategy. Given the lack of convergence in the literature to date, further studies characterizing valence asymmetries in learning using unconfounded measures will be needed to ascertain how broadly the biases we observed generalize to learning contexts with varying reward statistics (e.g., different outcome probabilities or outcomes that are truly negative instead of neutral).

Across two experimental samples, participants' idiosyncratic tendencies to place greater weight on outcomes that elicited either positive or negative prediction errors was, in turn, associated with a propensity to form stronger incidental memories for images paired with these outcomes during learning. This correspondence between valence biases in evaluation and in memory is consistent with past findings. Greater risk-seeking choice behavior has been associated with better memory for the magnitude of extreme win outcomes (*Ludvig et al., 2018*) as well as greater recalled frequency of win outcomes (*Madan et al., 2014*, *Madan et al., 2017*), whereas risk-averse choices have been associated with the opposite pattern. Our results extend these findings by linking individual risk preferences to an underlying learning algorithm that predicts the valence specificity of corresponding memory biases and by demonstrating that these biases extend to episodic features incidentally associated with valenced outcomes. Moreover, while several prior studies employing computational analyses of learning have variably observed enhanced memory for images coinciding with outcomes that elicit positive (*Davidow et al., 2016*; *Jang et al., 2019*), negative (*Kalbe and Schwabe, 2020*), or high-magnitude (independent of valence) PEs (*Rouhani et al., 2018*; *Rouhani and Niv, 2019*), our findings suggest that consideration of individual differences in the prioritization of positive versus negative PEs may be critical in understanding how these aspects of value-based learning signals relate to memory encoding.

Attention likely played a critical role in the observed learning and memory effects. Although our study did not include direct measures of attention, there is a large literature demonstrating the critical role of attention in both RL (*Dayan et al., 2000*; *Holland and Schiffino, 2016*; *Pearce and Hall, 1980*; *Radulescu et al., 2019*) and memory formation (*Chun and Turk-Browne, 2007*). Prominent theoretical accounts have proposed that attention should be preferentially allocated to stimuli that are more uncertain (*Dayan et al., 2000*; *Pearce and Hall, 1980*). In our study, memory was better for items that coincided with probabilistic compared to deterministic outcomes. This finding likely reflects greater attention to the episodic features associated with outcomes of uncertain choice options. Importantly, however, our memory findings could not be solely explained via an uncertainty-driven attention account as the relation between idiosyncratic asymmetric valence biases and memory was also evident within the subset of trials with probabilistic outcomes. Thus, our observed memory effects may reflect differential attention to valenced outcomes that varies systematically across individuals in a manner that can be accounted for by asymmetries in their learning rates. Such valence biases in attention have been widely observed in clinical disorders (*Bar-Haim et al., 2007*; *Mogg and Bradley, 2016*) and may also be individually variable within non-clinical populations.

In Experiment 1 of the present study, participants observed the outcomes of both free and forced choices. Prior studies have demonstrated differential effects of free versus forced choices on both learning and memory (*Chambon et al., 2020*; *Cockburn et al., 2014*; *Katzman and Hartley, 2020*), which may reflect greater allocation of attention to contexts in which individuals have agency. Valence asymmetries in learning have been found to vary as a function of whether choices are free or forced, such that participants tend to exhibit a greater positive learning rate bias for free than for forced

choices (*Chambon et al., 2020*; *Cockburn et al., 2014*). Here, we did not observe positive learning rate asymmetries for free choices, and a model that included separate valenced learning rates for free versus forced choices was not favored by model comparison. Studies have also found that subsequent memory is facilitated for images associated with free, relative to forced, choices (*Katzman and Hartley, 2020*; *Murty et al., 2015*). In Experiment 1, there was no significant effect of agency on memory. However, in Experiment 2, in which participants provided explicit predictions of choice outcomes, but did not make free choices, the qualitative pattern of learning and memory biases differed from that observed in Experiment 1, and closely resembled the pattern present within the subset of forced-choice trials from that experiment. Namely, in each of these conditions where participants were not able to make free choices, all participants, regardless of AI, exhibited better memory for images presented with large positive PEs. Thus, while our study was not explicitly designed to test for such effects, this preliminary evidence suggests that choice agency may modulate the relation between valence biases in learning and corresponding biases in long-term memory, a hypothesis that should be directly assessed in future studies.

While one interpretation of our results is that asymmetric value updating influences the prioritization of events in memory, recent theoretical proposals (*Biderman et al., 2020*; *Lengyel and Dayan, 2008*; *Shadlen and Shohamy, 2016*) and empirical findings (*Bakkour et al., 2019*; *Bornstein et al., 2017*; *Duncan et al., 2019*) suggest a potential alternative account. According to this work, sampling of specific valenced episodes from memory can influence decision-making and serve as a different way of making choices under uncertainty than the sort of incremental value computation formalized in RL models. Under this conceptualization, a tendency to preferentially encode or retrieve past positive or negative experiences may, in turn, drive risk-averse or risk-seeking choice biases. While our task design does not enable clear arbitration between these alternative directional hypotheses, our results provide additional evidence of a tight coupling between valuation and episodic memory, and further underscore the importance in examining individual differences in valence asymmetries in these processes.

Traditional behavioral economic models of choice suggest that risk preferences stem from a nonlinear transformation of objective value into subjective utility (*Bernoulli, 1954*; *Kahneman and Tversky, 1979*), with decreases in the marginal utility produced by each unit of objective value (i.e., a concave utility curve) producing risk aversion. Our present study was motivated by the insight that such risk-averse, or risk-seeking, preferences can also arise from an RL process that asymmetrically integrates valenced prediction errors (*Mihatsch and Neuneier, 2002*; *Niv et al., 2012*). In Experiment 1, we fit both a traditional behavioral economic model with exponential subjective utilities as well as a model with valenced learning rates. Notably, there was a very close correspondence between learning asymmetries derived from the valenced learning rate model and the risk preference parameter from the utility model, and model comparison indicated that these models provided comparably good accounts of participants' choice data. Thus, future research will be needed to arbitrate between utility and valenced learning rate models of decisions under risk. However, a potential parsimonious account is that a risk-sensitive learning algorithm could represent a biologically plausible process for the construction of risk preferences (*Dabney et al., 2020*), in which distortions of value are produced through differential subjective weighting of good and bad choice outcomes (*Mihatsch and Neuneier, 2002*; *Niv et al., 2012*).

Contrary to our a priori hypothesis, and to epidemiological (*Kann et al., 2018*; *Steinberg, 2013*) and theoretical (*Casey et al., 2008*; *Luna, 2009*; *Steinberg, 2008*) work suggesting that adolescence is a period of increased risk taking, we found that adolescents took fewer risks than children or adults in our task. While at first glance these results might appear anomalous, within the same sample, we found that adolescents reported greater real-world risk taking than both children and adults. This lack of correspondence between task-based and self-reported indices of risk taking is consistent with previous findings in adults (*Radulescu et al., 2020*), and suggests that these two measures reflect separable constructs. Past empirical studies assessing developmental changes in risky choice in laboratory tasks have observed varied results (*Defoe et al., 2015*; *Rosenbaum et al., 2018*; *Rosenbaum and Hartley, 2019*), but highlight two potential features of tasks that may elicit heightened adolescent risk taking. Adolescents may be more likely to take risks in tasks that require learning about risk through experience versus explicit description (*Rosenbaum et al., 2018*), and in which the probabilistic negative outcomes are rare (e.g., the Iowa Gambling Task; *Bechara et al., 1997*), qualities that

are also true of many real-world risk-taking contexts. While our task involved experiential learning, risky choices resulted in rewarding outcomes on half of the trials and non-win outcomes on the other half. Thus, undesirable outcomes were not rare and there were no true negative outcomes. Highlighting the important influence of such contextual features on decision-making across development, a recent study found that adolescents were more prone than adults to 'underweight' rare outcomes in decision-making relative to their true probabilities, conferring a greater propensity to take risks in situations where rare outcomes are unfavorable (*Rosenbaum et al., 2021*). Collectively, these findings suggest that specific details of an experimental design may influence the age-related patterns of risk taking observed in laboratory tasks (*Rosenbaum and Hartley, 2019*) and suggest that greater ecological validity of task designs might be best achieved by mirroring the key statistical properties of real-world decision contexts of interest.

The present findings raise the suggestion that, for a given individual, valence asymmetries in value-based learning might become more negative from childhood into adolescence, and more positive from adolescence into young adulthood. However, an important caveat is that such patterns of developmental change cannot be validly inferred from cross-sectional studies, which are confounded by potential effects of cohort (*Schaie, 1965*). Past studies have demonstrated that valence asymmetries in RL are indeed malleable within a given individual, exhibiting sensitivity to the statistics of the learning environment (e.g., the informativeness of positive versus negative outcomes; *Pulcu and Browning, 2017*) as well as to endogenous manipulations such as the pharmacological manipulation of neuromodulatory systems (*Michely et al., 2020*). Future longitudinal studies will be needed to definitively establish whether valence biases in learning exhibit systematic age-related changes within an individual over developmental time.

Adolescence is conventionally viewed as a period of heightened reward-seeking, begging the question of why adolescents might exhibit the strongest negative valence bias in learning and memory. Theoretical consideration of the adaptive role of valence asymmetries may provide a parsimonious resolution to this apparent contradiction (*Cazé and van der Meer, 2013*). Somewhat counterintuitively, greater updating for negative versus positive prediction errors (i.e., a negative valence bias) yields systematic distortions in subjective value that effectively increase the contrast between outcomes in the reward domain (e.g., a participant with a negative learning asymmetry will represent the risky 80- and 40-point machines as being more different from each other than a participant with a positive learning asymmetry), facilitating optimal reward-motivated action selection. This tuning of learning rates is particularly beneficial in environments in which potential rewards are abundant (*Cazé and van der Meer, 2013*), which may be true during adolescence when social elements of the environment acquire unique reward value (*Blakemore, 2008*; *Nardou et al., 2019*). While negative valence biases may be adaptive for reward-guided decision-making, a propensity to form more persistent memories for negative outcomes may also contribute to adolescents' heightened vulnerability to psychopathology (*Lee et al., 2014*; *Paus et al., 2008*). A recent study using computational formalizations found that adults who were biased toward remembering images associated with negative, relative to positive, prediction errors also exhibited greater depressive symptoms (*Rouhani and Niv, 2019*). Moreover, negative biases in real-world autobiographical memory are a hallmark of depression and anxiety in both adolescents (*Kuyken and Howell, 2006*) and adults (*Dillon and Pizzagalli, 2018*; *Gaddy and Ingram, 2014*). Future research should examine how valence biases in learning and memory, as well as the reward statistics of an individual's real-world environment, relate to vulnerability or resilience to psychopathology across adolescent development. Finally, given an extensive literature demonstrating the pronounced influence of neuromodulatory systems on both valence biases in RL (*Cox et al., 2015*; *Frank et al., 2004*; *Frank et al., 2007*; *Michely et al., 2020*) and value-guided memory (*Lisman and Grace, 2005*; *Sara, 2009*), future studies might examine how developmental changes within these systems relate to the age-related shifts in valence biases observed here.

## Materials and methods
### Experiment 1
#### Participants
Sixty-two participants ages 8–27 years were included in our final sample (mean age = 17.63, SD = 5.76, 32 females). Nine additional participants completed the study but were removed from the

sample due to poor task performance (described further below). This sample size is consistent with prior studies that used age as a continuous predictor and have found significant age differences in decision-making (e.g., *Decker et al., 2015*; *Potter et al., 2017*; *van den Bos et al., 2015*). All participants had no previous diagnosis of a learning disorder, no current psychiatric medication use, and normal color vision according to self- or parent report.

## Risk-sensitive RL task

In the present study, participants completed a risk-sensitive RL task adapted from *Niv et al., 2012* in which participants learned, through trial and error, the values and probabilities associated with five 'point machines' (*Figure 1A*). Three machines were deterministic and gave their respective payoffs 100% of the time (*Figure 1B*). Two machines were probabilistic (or risky) and gave their respective payoffs 50% of the time and zero points the other 50% (*Figure 1B*). Importantly, EV could be deconfounded from risk as there were two pairs of machines in which both probabilistic and deterministic options yielded the same EV (i.e., 100% 20 points and 50/50% 0/40 points; 100% 40 points and 50/50% 0/80 points). We presented each choice outcome on a 'ticket' that also displayed a trial-unique picture of an object. A subsequent memory test allowed us to explore the interaction between choice outcomes and memory encoding across age. The task was programmed in MATLAB Version R2017a (The MathWorks, Inc, Natick, MA).

All participants completed a tutorial that involved detailed instructions and practice trials with machines that had the same probability structure as the machines they would encounter in the later task (i.e., one machine always gave 1 point, the other gave 0 point on half of trials and 2 points on the other half). Then, participants completed the RL task, which included 183 trials. There were 66 total 'risky' choices between probabilistic and deterministic machines. 42 of these risky trials involved choices between machines with equal EV, while 24 trials required choices between the probabilistic 0/80 machine and the deterministic 20 point machine. Participants also experienced 75 single-option 'forced' trials (15 for each of the five machines) to ensure each participant learned about values and probabilities associated with all of the machines. During forced trials, only one machine appeared on the screen, and the participant pressed a button to indicate the location of the machine (left or right). Finally, there were 42 test trials in which one machine's value had absolute dominance over the other, with all outcomes of one option being greater than or equal to all outcomes of the other option (e.g., 100% chance of 40 points versus 50% of 40 points and 50% of 0 points). Test trials allowed us to gauge participants' learning and understanding of the task. We excluded nine participants who did not choose correctly on at least 60% of test trials in the last 2/3 of the task (four children ages 8–9, three adolescents ages 14–16, and two adults ages 24–25). The trials were divided into blocks with 1/3 of the trials in each block, and after each block, participants could choose to take a short break. Unbeknownst to participants, trials were pseudo-randomized, such that 1/3 of each type of trial was presented in each block of the task, with the order of trial types randomized within each block. Outcomes of risky machines were additionally pseudo-randomized so that within each series of eight choices from a given risky machine, four choices resulted in a win and four resulted in zero point, in a random order.

On each trial, participants were asked to make a choice within 3 s after the machines were presented. If they chose in time, the outcome of the choice was presented on a 'ticket' along with a randomly selected, trial-unique picture of an object for 2 s (*Figure 1A*). If they did not respond in time, the words 'TOO SLOW' were presented, without a picture, for 1 s before the task moved to the next trial. Across all participants, only 37 (out of 11,346) total trials were missed for slow responses, with a maximum of 7 missed trials for one participant.

After completing the choice task, participants were probed for their explicit memory of points associated with each machine. For every machine, a participant was first asked, "Did this machine always give you the same number of points, or did it sometimes give 0 points and sometimes give you more points?" If the participant indicated that the machine always gave the same number of points, they were asked, "How many points did this machine give you each time you chose it?" Otherwise, they were asked, "How many points did this machine give you when it did not give 0 points?" To respond to this second question, participants selected from all possible point outcomes presented in the task (0, 20, 40, 80). There was no time limit for responding to these questions.

Next, participants completed a surprise memory test, in which all 183 images presented during the task and 183 novel images were presented in random order (*Figure 1C*). Images corresponding to the few choice trials that were missed due to slow responses were recategorized as novel. Ratings were on a scale from 1 (definitely saw during the task) to 4 (definitely did not see during the task), and participants had unlimited time to indicate their responses. All images were obtained from the Bank of Standardized Stimuli (BOSS; *Brodeur et al., 2010*; *Brodeur et al., 2014*) and were selected to be familiar and nameable for the age range in our sample. For each participant, half of the set of photos was randomly chosen to be presented during the task and half were assigned to be novel images for the memory test.

## Self-reported risk taking

To assess the predictive validity of our findings for real-world risk taking, participants completed the DOSPERT scale (*Blais and Weber, 2006*). The DOSPERT indexes participants' likelihood of taking risks in five domains: monetary, health and safety, recreational, ethical, and social. We computed the mean self-reported likelihood of risk taking across all behaviors on the DOSPERT as a measure of real-world risk taking. Age-appropriate variants of the DOSPERT were administered to children (8–12 years old), adolescents (13–17 years old), and adults (ages 18 and older) (*Barkley-Levenson et al., 2013*; *Somerville et al., 2017*; *van Duijvenvoorde et al., 2016*).

## Reasoning assessment

We administered the Vocabulary and Matrix Reasoning sections of the Wechsler Abbreviated Scale of Intelligence (WASI; *Wechsler, 2011*), which index verbal cognition and abstract reasoning, to ensure that these measures were not confounded with age within our sample. WASI scores did not vary by linear or quadratic age (*p*s>.2). Thus, we did not include this measure in subsequent analyses.

## Procedure

Participants first provided informed consent (adults) or assent and parental consent (children and adolescents). Next, participants completed the risk-sensitive RL task and memory test, followed by the DOSPERT questionnaire and the WASI. Participants were paid $15 for completing the experiment, which lasted approximately 1 hr. Although participants were told that an additional bonus payment would be based on the number of points they earned in the risk-sensitive RL task, all participants received the same $5 bonus payment. The study protocol was approved by the New York University Institutional Review Board.

## Analyses

### Reinforcement-learning models

Four RL models were fit to participants' choices in the task.

#### TD model

We fit a TD learning model (*Sutton and Barto, 1998*), in which the estimated value of choosing a given machine ($Q_M$) is updated on each trial ($t$) according to the following function: $Q_M(t + 1) = Q_M(t) + \alpha * \delta(t)$, in which $\delta(t) = r(t) – Q_M(t)$ is the prediction error, representing how much better or worse the reward outcome ($r$) is than the estimated value of that machine. $\delta$ is scaled by a learning rate $\alpha$, a free parameter that is estimated separately for each participant.

#### RSTD model

The RSTD model is similar to the TD model but includes two separate learning rates for prediction errors of different signs. Specifically, when $\delta$ is positive, the value of the chosen machine is updated according to the equation: $Q_M(t + 1) = Q_M(t) + \alpha^+ * \delta(t)$. When $\delta$ is negative, the chosen machine's value is updated as $Q_M(t + 1) = Q_M(t) + \alpha^- * \delta(t)$. Including two learning rates allows the model to be sensitive to the risk preferences revealed by participants' choices across the probabilistic and deterministic ('risky versus safe') choice pairs in the paradigm (*Niv et al., 2012*). For a given individual, if $\alpha^+$ is greater than $\alpha^-$, Q-values of the machines with variable outcomes will be greater than those of deterministic machines with equal EV, and the individual will be more likely to make risk-seeking choices. Conversely if $\alpha^-$ is greater than $\alpha^+$, the Q-values of the risky machines will be lower than their EVs, making risk-averse choices more likely. To index the relative difference between $\alpha^+$ and $\alpha^-$, we

computed an AI as AI = $(\alpha^+ - \alpha^-)/(\alpha^+ + \alpha^-)$, where an AI > 0 reflects greater weighting of positive relative to negative prediction errors, whereas an AI < 0 reflects greater relative weighting of negative prediction errors (**Niv et al., 2012**).

### FourLR model

In our task, participants made both free and forced choices. Past research suggests that valence biases in learning may differ as a function of choice agency (**Chambon et al., 2020**; **Cockburn et al., 2014**). To test this possibility, we assessed the fit of a FourLR model, which was the same as the RSTD model except that it included four learning rates instead of two, with separate $\alpha^+$ and $\alpha^-$ parameters for free and forced choices.

### Utility model

As a further point of comparison with the TD, RSTD, and FourLR models, we estimated a utility model that employed the same value update equation as the TD model, $Q_M(t + 1) = Q_M(t) + \alpha * \delta(t)$. However, $\delta$ was defined according to the equation $\delta(t) = r(t)^\rho - Q_M(t)$, in which the reward outcome is exponentially transformed by $\rho$, which represents the curvature of each individual's subjective utility function (**Pratt, 1964**). $\rho < 1$ corresponds to a concave utility function, which yields risk aversion as a result of diminishing sensitivity to returns (**Tversky and Kahneman, 1992**). In contrast, $\rho > 1$ corresponds to a convex utility function that yields risk-seeking behavior.

In all models, Q-values were converted to probabilities of choosing each option in a trial using the softmax rule, $P_{M1} = e^{\beta*Q(t)M1}/(e^{\beta*Q(t)M1} + e^{\beta*Q(t)M2})$, where $P_{M1}$ is the predicted probability of choosing Machine 1, with the inverse temperature parameter $\beta$ capturing how sensitive an individual's choices are to the difference in value between the two machines. Notably, outcomes of the forced trials were included in the value updating step for each model. However, forced trials were not included in the modeling stage in which learned values are passed through the softmax function to determine choice probabilities as there was only a single-choice option on these trials.

## Model fitting

Prior to model fitting, outcome values were rescaled between 0 and 1, with 1 representing the maximum possible point outcome (80). We fit all RL models for each participant via maximum a posteriori estimation in MATLAB using the optimization function fminunc. Q-values were initialized at 0.5 (equivalent to 40 points). Bounds and priors for each of the parameters are listed in **Table 1**. There was no linear or quadratic relationship between BIC and age in any of the models (all $p$s>0.1).

**Table 1.** Bounds, priors, and recoverability for parameters in each model.

| Model | Parameter | Bounds | Prior | Recoverability |
|---|---|---|---|---|
| TD | $\alpha$ | 0,1 | Beta(2,2) | 0.84 |
| | $\beta$ | 0.000001, 30 | Gamma(2,3) | 0.88 |
| RSTD | $\alpha^+$ | 0,1 | Beta(2,2) | 0.79 |
| | $\alpha^-$ | 0,1 | Beta(2,2) | 0.88 |
| | $\beta$ | 0.000001, 30 | Gamma(2,3) | 0.90 |
| FourLR | $\alpha^+$ free | 0,1 | Beta(2,2) | 0.79 |
| | $\alpha^-$ free | 0,1 | Beta(2,2) | 0.89 |
| | $\alpha^+$ forced | 0,1 | Beta(2,2) | 0.76 |
| | $\alpha^-$ forced | 0,1 | Beta(2,2) | 0.78 |
| | $\beta$ | 0.000001, 30 | Gamma(2,3) | 0.90 |
| Utility | $\alpha$ | 0,1 | Beta(2,2) | 0.75 |
| | $\beta$ | 0.000001, 30 | Gamma(2,3) | 0.88 |
| | $\rho$ | 0, 2.5 | Gamma(1.5,1.5) | 0.88 |

Priors for $\alpha$ and $\beta$ were based on those used in **Niv et al., 2012**.

TD, temporal difference; RSTD, risk-sensitive temporal difference; LR, learning rate.

**Table 2.** Model recovery.

| | | Comparison model | | | |
|---|---|---|---|---|---|
| | | **TD** | **RSTD** | **FourLR** | **Utility** |
| | TD | - | 0.98 | 1.00 | 0.97 |
| | RSTD | 0.57 | - | 0.99 | 0.65 |
| | FourLR | 0.50 | 0.31 | - | 0.39 |
| Generating model | Utility | 0.58 | 0.76 | 0.99 | - |

TD, temporal difference; RSTD, risk-sensitive temporal difference; LR, learning rate.

## Parameter and model recovery

For each model, we simulated data for 10,000 subjects with values of each parameter drawn randomly and uniformly from the range of possible parameter values. Next, we fit the simulated data using the same model. We tested for recoverability of model parameters by correlating the parameter that generated the data with the parameters produced through model fitting. These correlations are displayed in *Table 1*. All parameters for TD, RSTD, FourLR, and Utility models showed high recoverability.

To examine the identifiability of the TD, RSTD, FourLR, and Utility models, we generated simulated data using each model and fit all four of the models, including those that were *not* used to generate the data to each simulated dataset (e.g., we fit all the TD-generated subjects with the TD model as well as the RSTD, FourLR, and Utility models). We then used BIC, a quality-of-fit metric that penalizes models for additional parameters, to assess whether the generating model was also the best-fitting model for each subject. Recoverability was reasonable for all models except the least-parsimonious FourLR model (*Table 2*). Aside from the subjects generated by the FourLR model, for all pairwise comparisons between generating and comparison models, the majority of simulated subjects were best fit by the generating model. The RSTD-simulated subjects who were better fit by the TD model were those who had less extreme AI values (*Appendix 1—figure 7*), and thus could be more parsimoniously captured by a model with a single learning rate. We also found that RSTD model parameters were reasonably well recovered across the range of AI observed in our empirical sample (see *Appendix 1—figure 8*).

Values in this table indicate the proportion of participants simulated by the generating model who are best fit by the generating model in a pairwise comparison with each alternative model.

## Statistical analyses

Statistical analyses were performed in R version 4.0.2 (*R Development Core Team, 2016*) with a two-tailed alpha threshold of $p < 0.05$. For tests of trial-wise effects, we ran linear mixed-effects regression (lmer) or generalized linear mixed-effects regression (glmer) models (lme4 package; *Bates et al., 2015*), which included participant as a random effect, and estimated random intercepts and slopes for each fixed effect. We used the 'bobyqa' optimizer with 1 million iterations. Trial number was included in lmer and glmer regression models. All independent variables were z-scored. We began with this maximal model, which converged for all analyses except one, for which we systematically reduced the complexity of the model until it converged (*Barr et al., 2013*; see Appendix 1). In RT analyses, we removed responses that were less than 0.2 s (n = 22, out of 11,309 total trials, with a maximum of 9 for one participant) and log-transformed RT prior to running regressions. To test for linear effects of age, we included z-scored age in regression models. Potential quadratic age effects were assessed by adding a squared z-scored age term in a regression model. We used the anova function to arbitrate between these regression models and report only linear age effects if the addition of quadratic age did not significantly improve model fit. To probe whether a quadratic age effect qualifies as u-shaped, we used the two-lines approach (*Simonsohn, 2018*), which algorithmically determines a break point in the distribution and tests whether regression lines on either side of the break point have significant slopes with opposite signs.

## Reporting

For one-way and paired *t*-tests, we report *t*-statistics, p-values, and Cohen's d with 95% confidence intervals (CIs; using the function cohens_d in the rstatix package [one-way *t*-test] or the effectsize package [paired *t*-test]).

For linear regressions, we report unstandardized regression coefficients, *t*-statistics, and p-values. We also report Cohen's $f^2$ with 95% CIs, a standardized effect size measure (*Cohen, 1992*) computed by squaring the output of the function cohens_f in the effectsize package.

For multilevel models, we report test statistics (*t* for linear mixed-effects models and *z* for generalized linear mixed-effects models), p-values, and unstandardized effect sizes with 95% CIs (unstandardized coefficients for linear mixed-effects models, and odds ratios for generalized linear mixed-effects models).

## Experiment 2

Next, we looked for evidence that valence biases in learning influence memory in a previously published independent dataset (*Rouhani et al., 2018*). Notably, results from this study suggested that unsigned PEs (i.e., PEs of greater magnitude whether negative or positive) facilitate subsequent memory, but no signed effect was observed. Here, we examined whether signed valence-specific effects might be evident when we account for individual differences in learning.

Briefly, adult participants completed a Pavlovian learning task (i.e., participants did not make choices and could not influence the observed outcomes) in which they encountered indoor and outdoor scenes. One type of scene had higher average value than the other. On each trial, an indoor or outdoor image was displayed, and participants provided an explicit prediction for the average value of that scene type. After the learning task, participants completed a memory test for the scenes. A detailed description of the experimental paradigm can be found in the original publication (*Rouhani et al., 2018*).

In order to derive each participant's AI, we fit an 'Explicit Prediction' RL model to the participants' estimation data (see Appendix 1 for more details on our model specification and fitting procedure). Similar to our RSTD model, this model included separate learning rates for trials with positive and negative PEs.

Importantly, the RSTD model and the Explicit Prediction model differed in that the RSTD model included a β parameter, while the Explicit Prediction model did not. In Experiment 1, this extra parameter allowed us to use the softmax function to convert the relative estimated values of the two machines into a probability of choosing each machine presented, which we then compared to participants' actual choices during maximum likelihood estimation. In contrast, in Experiment 2, participants explicitly reported their value predictions (and did not make choices), so the model's free parameters were fit by minimizing the difference between the model's value estimates and participants' explicit predictions.

## Acknowledgements

We thank Nina Rouhani and Yael Niv for sharing their data and providing feedback on the analyses in Experiment 2, and Alexandra Cohen and Kate Nussenbaum for providing constructive feedback on the manuscript.

This work was funded by an NSF CAREER grant (1654393), an NIMH BRAINS grant (R01MH126183), the NYU Vulnerable Brain Project, and a Jacobs Foundation Early Career Award awarded to CAH, and by a National Institute on Drug Abuse NRSA awarded to GMR (F32DA047047).

## Additional information

### Funding

| Funder | Grant reference number | Author |
| --- | --- | --- |
| Jacobs Foundation | | Catherine Hartley |

| Funder | Grant reference number | Author |
|---|---|---|
| National Science Foundation | 1654393 | Catherine Hartley |
| NYU Vulnerable Brain Project | | Catherine Hartley |
| National Institute on Drug Abuse | F32DA047047 | Gail M Rosenbaum |
| National Institute of Mental Health | R01MH126183 | Catherine Hartley |

The funders had no role in study design, data collection and interpretation, or the decision to submit the work for publication.

### Author contributions
Gail M Rosenbaum, Conceptualization, Data curation, Formal analysis, Funding acquisition, Methodology, Software, Supervision, Validation, Visualization, Writing – original draft, Writing – review and editing; Hannah L Grassie, Data curation, Formal analysis, Investigation, Project administration, Writing – original draft; Catherine A Hartley, Conceptualization, Funding acquisition, Methodology, Resources, Supervision, Writing – original draft, Writing – review and editing

### Author ORCIDs
Gail M Rosenbaum  http://orcid.org/0000-0002-6306-0508
Catherine A Hartley  http://orcid.org/0000-0003-0177-7295

### Ethics
Human subjects: Participants provided informed consent (adults) or assent and parental consent (children and adolescents). The study protocol was approved by the New York University Institutional Review Board (IRB#2016-1194).

### Decision letter and Author response
Decision letter https://doi.org/10.7554/eLife.64620.sa1
Author response https://doi.org/10.7554/eLife.64620.sa2

---

# Additional files

### Supplementary files
- Transparent reporting form
- Supplementary file 1. Results from an ordinal model predicting memory performance.

### Data availability
Data and code are available on the Open Science Framework. Experiment 1 data were generated in the present study. Experiment 2 data are provided in our repository, but were collected as part of the following study: Rouhani, N., Norman, K. A., & Niv, Y. (2018). Dissociable effects of surprising rewards on learning and memory. Journal of Experimental Psychology: Learning, Memory, and Cognition, 44(9), 1430-1443. https://doi.org/10.1037/xlm0000518.

The following dataset was generated:

| Author(s) | Year | Dataset title | Dataset URL | Database and Identifier |
|---|---|---|---|---|
| Rosenbaum GM, Grassie HL, Hartley CA | 2020 | Valence biases in reinforcement learning shift across adolescence and modulate subsequent memory | https://osf.io/srtgc/ 10.17605/OSF.IO/ SRTGC | Open Science Framework, 10.17605/OSF.IO/SRTGC |

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

## Appendix 1

### Experiment 1
Learning and risk taking by age group and block

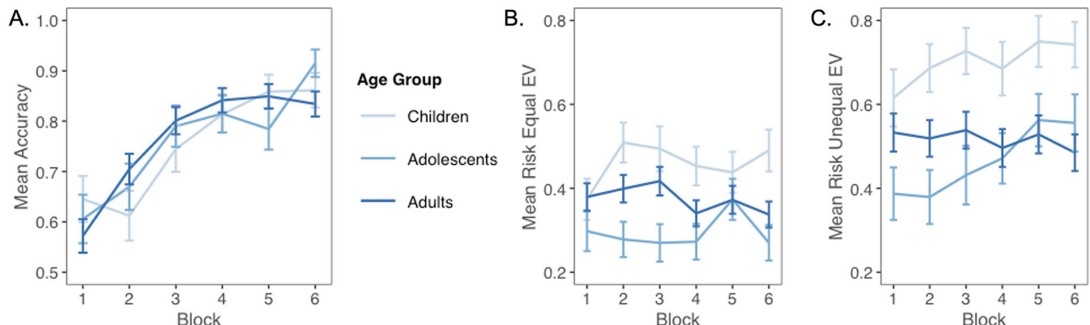

**Appendix 1—figure 1.** Choices as a function of block and age group. (**A**) Mean accuracy on test trials (approximately seven per participant per block). (**B**) Mean risk taking for equal expected value (EV) trials (seven per participant per block). (**C**) Mean risk taking for unequal-EV trials (four per participant per block).

### Unequal-EV choices of probabilistic machines

On trials where participants faced risky and safe choice options with unequal EV (i.e., the 0/80 point machine vs. the safe 20 point machine), age patterns were similar to those for equal-EV trials. Specifically, there was a significant quadratic age effect on probabilistic, risky decision-making ($b$ = 0.09, 95% CI [0.01, 0.16], $t(59)$ = 2.21, p=0.031, $f^2$ = 0.08, 95% CI [0, 0.30]). The two-lines test showed that unequal-EV risk taking significantly decreased from ages 8–17 ($b$ = 0.04, $z$ = –2.77, p=0.006), and marginally increased from ages 17–27 ($b$ = 0.03, $z$ = 1.74, p=0.081).

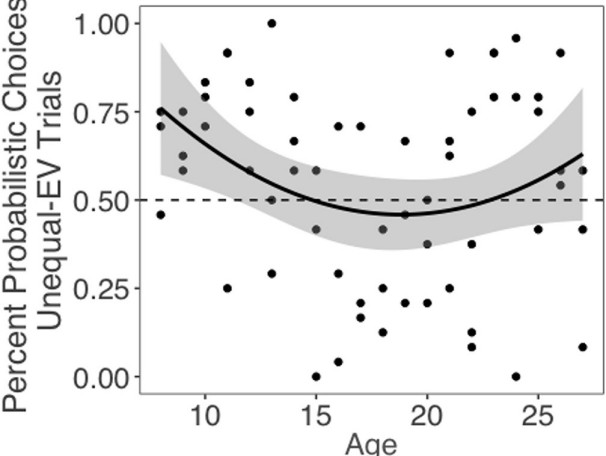

**Appendix 1—figure 2.** Probabilistic choices for unequal-expected value (EV) risk trials. Probabilistic (i.e., risky) choices by age on trials with unequal-expected value (EV) risky and safe machines, with a choice between the 0/80 probabilistic machine and the deterministic 20-point machine. Data points depict the mean percentage of trials where each participant selected the probabilistic choice option as a function of age. Regression line is from the glmer model including linear and quadratic age terms. Shaded region represents 95% CIs for estimates.

## Memory performance by age, PE valence, and asymmetry index

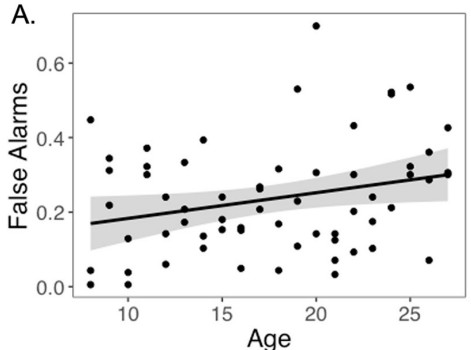
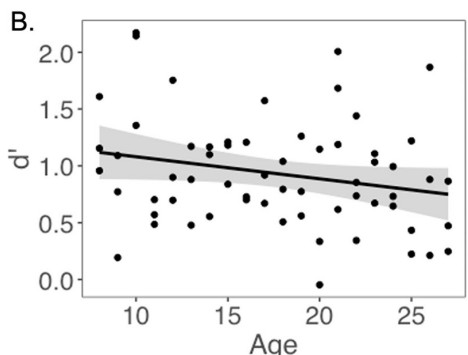

**Appendix 1—figure 3.** Memory performance across age. (**A**) False alarm rate as a function of age. As reported in the article, false alarm rate increased with age (p=0.037). (**B**) D' as a function of age. As reported in the article, there is a marginal linear decrease in d' with age (p=0.070).

## Multilevel model fitting

The maximal multilevel model did not converge when we tested for the three-way interaction between PE valence, PE magnitude, and AI predicting memory accuracy. We systematically reduced the model until we found a model that converged (***Barr et al., 2013***). The maximal model that converged is:

Memory Response ~ Age_Z + Age_Z^2+ Memory Trial Number + False Alarm Rate+ AI * PE Valence * PE Magnitude + (1+ PE Magnitude + Memory Trial Number || SubjectNumber)

## Ordinal modeling of memory data

Our multilevel models of memory data collapsed across confidence ratings (e.g., 'Definitely old' and 'Maybe old'), a convention widely adopted in manuscripts examining memory accuracy effects (e.g., ***Dunsmoor et al., 2015***; ***Murty et al., 2016***). As an exploratory analysis, we ran an ordinal model using the clmm function in the ordinal R package (***Christensen, 2019***), which allowed us to test for an AI × PE Valence × PE Magnitude interaction effect using participants' uncollapsed memory responses as the dependent variable (1 = definitely new, 2 = maybe new, 3 = maybe old, and 4 = definitely old).

Regression results are reported in Appendix 1—table 1 (see ***Supplementary file 1***). Importantly, the results from the ordinal regression were not meaningfully different from results collapsed across confidence ratings. In particular, the AI × PE Valence × PE Magnitude interaction was significant in the ordinal regression. The three-way interaction is plotted in ***Appendix 1—figure 4***. Here, probabilities of each memory response are plotted as a function of PE valence and magnitude separately for AI = –0.8 (top panels), AI = 0 (middle panels), and AI = 0.8 (bottom panels). Consistent with the results reported in the main text, the likelihood of a 'definitely old' response was highest for those in those with low AI for images that coincided with high-magnitude negative PEs (top-left panel) and those with high AIs for images that coincided with high-magnitude positive PEs (bottom-right panel).

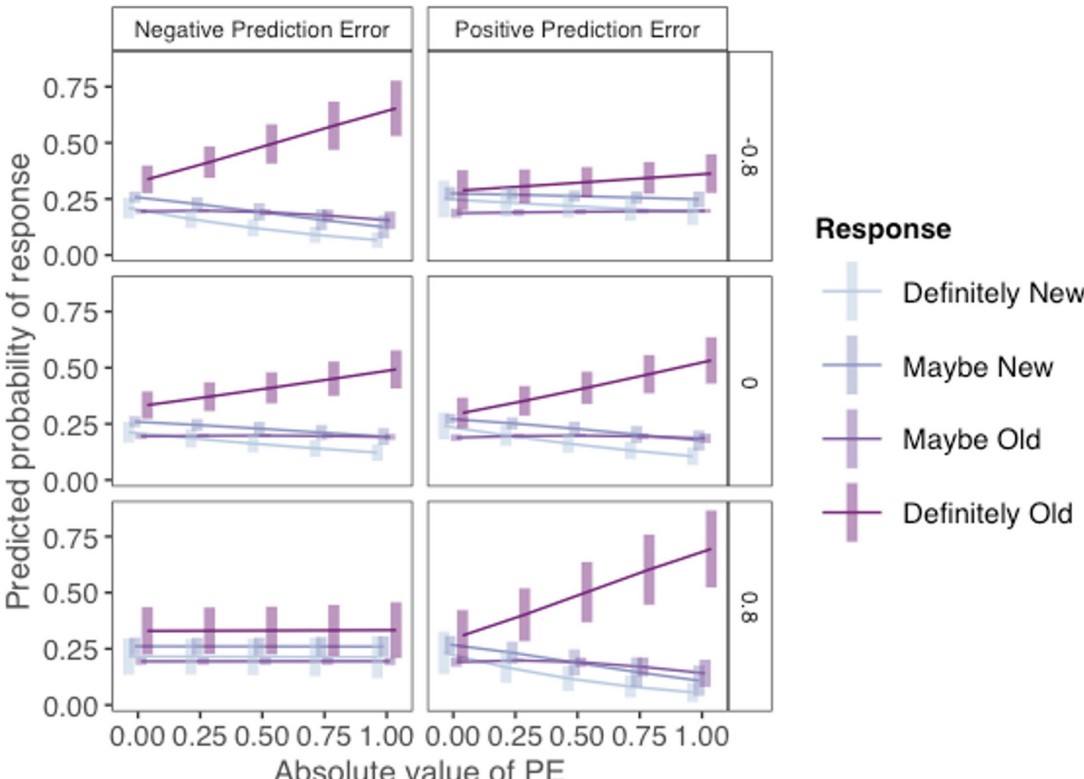

**Appendix 1—figure 4.** Ordinal regression analysis of incidental memory judgments (Experiment 1). Results from an ordinal regression demonstrating that incidental memory accuracy for pictures presented with choice outcomes varies as a function of PE valence, PE magnitude, and asymmetry index (AI) without collapsing across response confidence levels. The probability of each memory response is plotted separately for three different AI levels (top: AI = –0.8; middle: AI = 0; bottom: AI = 0.8) as a function of PE valence, PE magnitude.

## BIC distributions

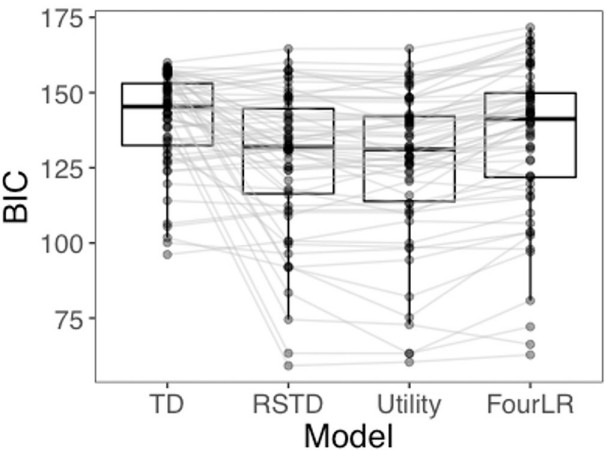

**Appendix 1—figure 5.** BIC distributions for all four models tested.

## RSTD vs. TD model fit as a function of asymmetry index

The RSTD model fit the data of subjects with more extreme AI values substantially better than the TD model (*Appendix 1—figure 6*). The difference in model fit was smaller for those with AIs close

to 0, reflecting the redundancy of a model including two learning rates when the learning rates are similar.

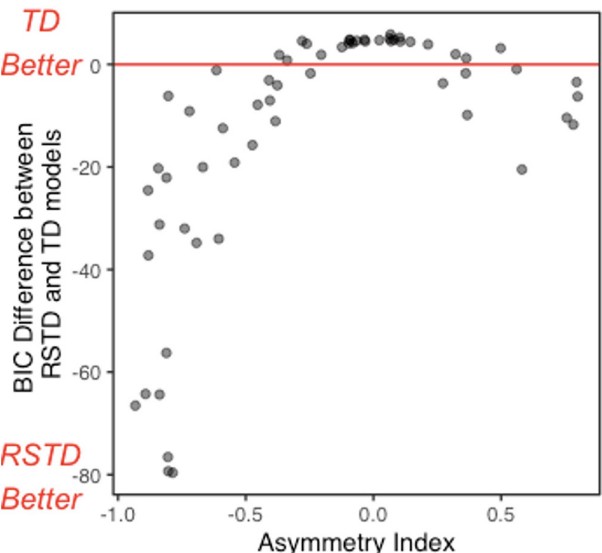

**Appendix 1—figure 6.** Relative BIC as a function of asymmetry index (AI). The difference between risk-sensitive temporal difference (RSTD) and temporal difference (TD) model fit (BIC) for all participants in Experiment 1. Values below 0 indicate a better fit by the RSTD model.

## RSTD vs. TD reinforcement learning model recovery

In the main text methods, we described our model recovery analysis, where we simulated 10,000 'subjects' using each model and fit the simulated data using the generating model, and all alternative models. *Appendix 1—figure 7* shows relative RSTD and TD model fit (BIC) model fit for subjects generated by the RSTD model, as a function of AI. For simulated subjects with a more extreme AI, the RSTD model provided a substantially better fit. For those with AIs closer to 0 (i.e., when $\alpha^+$ is similar to $\alpha^-$), RSTD and TD models perform similarly.

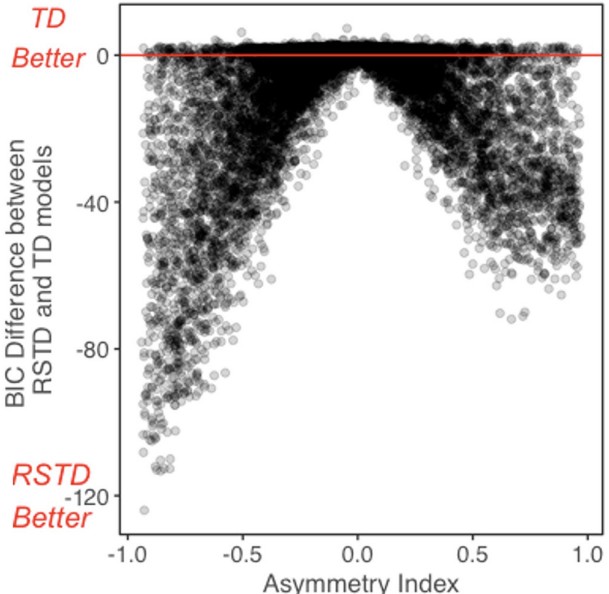

**Appendix 1—figure 7.** Relative BIC as a function of asymmetry index (AI) for participants simulated by the risk-sensitive temporal difference (RSTD) model. The difference between risk-sensitive
*Appendix 1—figure 7 continued on next page*

*Appendix 1—figure 7 continued*
temporal difference (RSTD) and temporal difference (TD) model fit (BIC). The difference in model fit (BIC) between the risk-sensitive temporal difference (RSTD) and temporal difference (TD) models for 10,000 subjects simulated using the RSTD model. Values below 0 indicate a better fit by the RSTD model.

## RSTD model recovery as a function of asymmetry index
We tested whether parameter recovery differed as a function of individual differences in the propensity to make deterministic choices. This question was of particular interest because those who tended to make deterministic choices were less likely to choose risky machines, and therefore were less likely to experience high-magnitude PEs (for a related discussion, see the 'Utility and subsequent memory' section below, along with *Appendix 1—figure 11F*). To this end, we tested whether parameter recovery and model fit (BIC) varied as a function of AI. Importantly, AI is highly correlated with risk taking, so this analysis allowed us to test for potential differences in parameter recoverability for participants who more frequently chose the deterministic point machines (i.e., those with low AI). To this end, we divided the simulated participants into AI quartiles and examined parameter recovery and model fit in each AI quartile. We found that the parameters were reasonably well recovered at all levels of AI (*Appendix 1—figure 8A–C*).

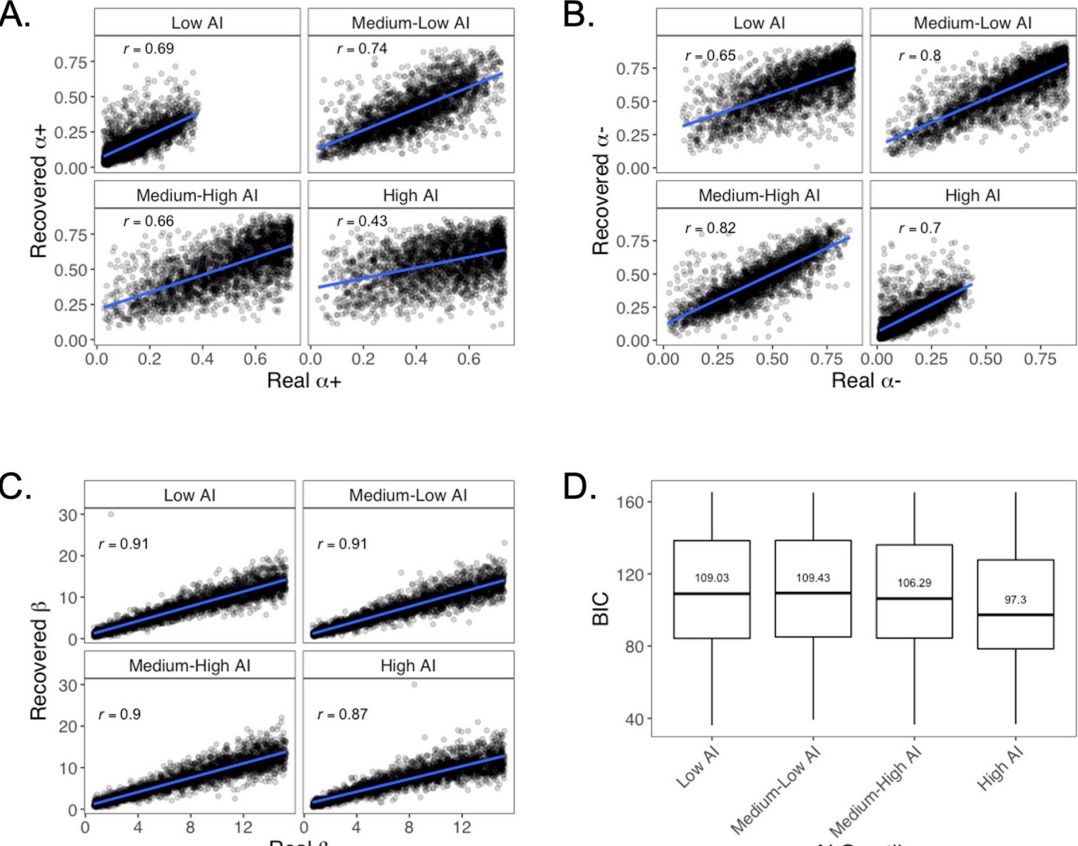

**Appendix 1—figure 8.** Parameter recovery at different levels of Asymmetry Index (AI). Parameter recovery for simulated participants at low (AIs ranging from –0.94 to –.0374), medium-low (AIs ranging from –0.373 to –0.07684), medium-high (AIs ranging from –0.07683 to 0.2501), and high (AIs ranging from 0.2502 to 0.97) levels of AI. (**A**) $\alpha^+$ recovery. (**B**) $\alpha^-$ recovery. (**C**) $\beta$ recovery. (**D**) BIC.

However, parameter recoverability varied across levels of AI, somewhat counterintuitively. In particular, recovery of the $\alpha+$ parameter was relatively poorer for the simulated participants in the high-AI quartile (*Appendix 1—figure 8A*) and $\alpha-$ recoverability was relatively poorer for those in the low-AI quartile (*Appendix 1—figure 8B*). Taken together, these patterns suggest that learning

rate parameters are relatively less well recovered for individuals with higher AIs (i.e., who made more risk-seeking choices).

This differential recoverability as a function of AI stems from the interactions between subjects' risk preferences and the set of risky choice trials presented in our task. There were two types of risky trials in our task: equal-EV (0/40 vs. 20, or 0/80 vs. 40) and unequal-EV (0/80 vs. 20). This particular combination of equal- and unequal-EV risk trials led to differential resolution in the estimation of valenced learning rates as a function of AI. Positive learning rates for risk-averse participants could be estimated more accurately because those who were very risk averse (and thus had a much larger $\alpha^-$ than $\alpha^+$) might choose both the safe 40-point option and the safe 20-point option over the 0/80 machine, whereas those who were less risk averse might prefer the safe 40 to the 0/80, but the 0/80 over the safe 20. In contrast, those with high positive AI are likely to choose the risky option on every equal and unequal-EV trial, so the ability to distinguish precisely between different levels of $\alpha+$ for those with high AI is diminished. Our model fit results provide further evidence that this lower $\alpha+$ recoverability in individuals with high AIs stems from this aspect of our task structure (*Appendix 1—figure 8D*). Despite the model's imprecise $\alpha^+$ estimation for these high-AI subjects, the model was able to predict behavior well (i.e., BIC was low), likely because these participants are likely to take risks on all equal and unequal-EV risk trials, regardless of their precise $\alpha^+$ level.

This increase in parameter recovery for those participants with low compared to high AI runs counter to the notion that smaller PEs may drive worse recovery in these low-AI participants. Although it is true that PEs were smaller for those with relatively lower AIs (see *Appendix 1—figure 11F*), our design required these low-AI participants to experience high-magnitude PEs from risky choices on some trials. Specifically, in our task, participants encountered forced trials, where participants were required to choose specific machines, which were sometimes risky, and test trials (where one option dominated the other), and the dominating option was sometimes risky. Thus, including these forced and test trials may have facilitated our ability to recover learning rates in those with low AI, while also providing opportunities for participants to sufficiently learn and demonstrate their knowledge of machine outcomes and probabilities.

Despite these differences in parameter recoverability at different levels of AI within this experimental paradigm, there are several reasons why we do not believe that these results are problematic for interpreting the current results. First, these simulations were generated by sampling uniformly from the range of learning rate and temperature parameters observed in our empirical sample. Thus, these simulated participants can take on AI levels that are not actually represented within our empirical sample. Indeed, 81% of fit AI values observed in our empirical sample fall within the lower three quartiles of AI values for this simulation, for which parameter recoverability was higher. Moreover, the recoverability estimates in this simulation dramatically overrepresent participants with low levels of decision noise relative to our empirical sample, which distorts these estimates of recoverability to be lower than what would actually be obtained for our empirical sample of participants (e.g., when we exclude any simulated participants with decision noise <2, which captures the vast majority of participants in our sample, recoverability of *r* values all increase by ~.05).

### Posterior predictive check

We ran a posterior predictive check on both RSTD and Utility models. We simulated 100 subjects for every real participant's empirically derived parameters, using both the Utility model and the RSTD model, and took the mean proportion of risks across all 100 simulated subjects for each model. The quadratic age pattern in (simulated) risk taking for equal EV trials was significant for both RSTD- and Utility-simulated data (RSTD: *Appendix 1—figure 9B*, *b* = 0.05, 95% CI [0, 0.10], *t*(59) = 2.16, p=0.035, $f^2$ = 0.08, 95% CI [0, 0.29]; Utility: *Appendix 1—figure 9D*, *b* = 0.05, 95% CI [0.01, 0.10], *t*(59) = 2.29, p=0.026, $f^2$ = 0.09, 95% CI [0, 0.31]). Additionally, choices derived from both RSTD and Utility model simulations using each participant's best-fit parameter estimates were highly correlated with actual choices (RSTD: *r* = 0.92, 95% CI [0.87, 0.95], *t*(60) = 18.14, p<0.001; Utility: *r* = 0.89, 95% CI [0.83, 0.93], *t*(60) = 15.25, p<.001). However, the correlation between simulated and actual choices was significantly stronger for participants simulated using the RSTD model compared to the Utility model (*t*(61) = 2.58, p<0.012). Thus, it appears that the RSTD model provides a better qualitative fit to the choice data compared to the Utility model.

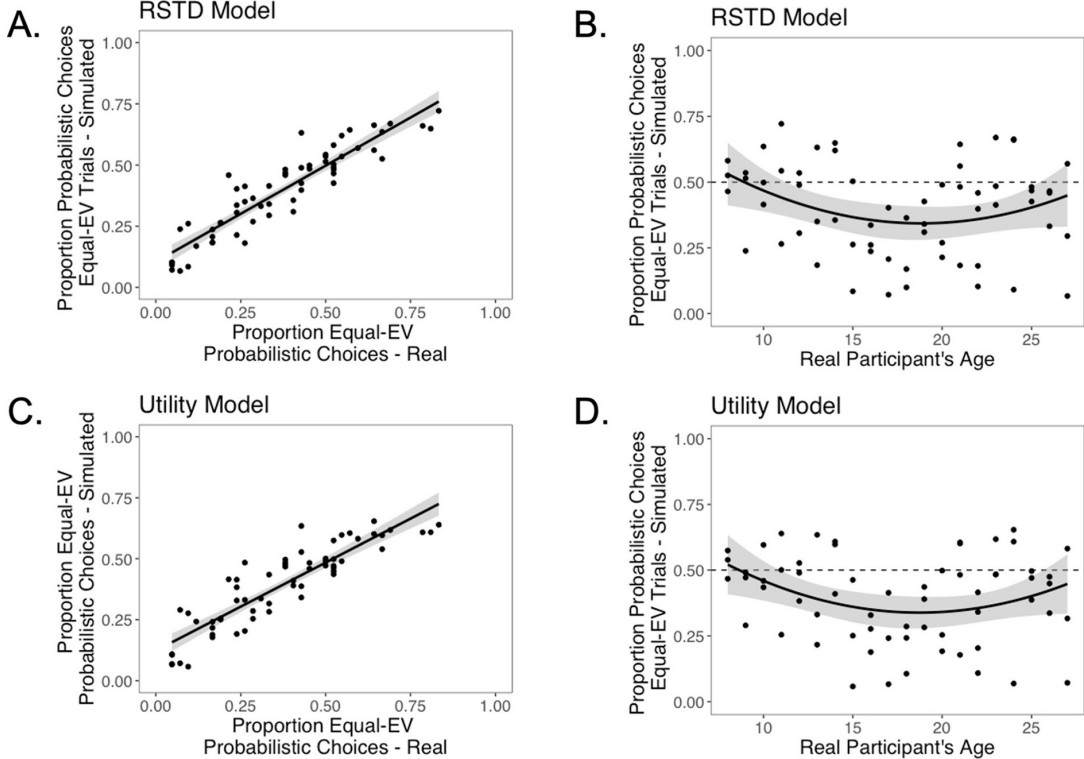

**Appendix 1—figure 9.** Posterior prediction check results. Relationship between choices of the risky (probabilistic) option in real versus simulated data for (**A**) the risk-sensitive temporal difference (RSTD) model and (**C**) the Utility model. Relationship between risky choices and the corresponding real participant age for (**B**) RSTD-simulated and (**D**) Utility-simulated participants.

## Age patterns in RSTD parameters

As reported in the main text, the relation between AI and age appears to be driven by quadratic age patterns in $\alpha^-$ ($b$ = –0.09, 95% CI [–0.15, –0.03], $t$(59) = –3.01, p=0.004, $f^2$ = 0.15, 95% CI [0.02, 0.43]; **Appendix 1—figure 10B**). The relation between linear age and $\alpha^+$ was not significant ($b$ = –0.02, 95% CI [–0.07, 0.03], $t$(60) = –0.85, p=0.401, $f^2$ = 0.01, 95% CI [0, 0.13]; **Appendix 1—figure 10A**). Additionally, the relation between age and $\beta$ was not significant ($b$ = 0.56, 95% CI [–0.22, 1.33], $t$(60) = 1.44, p=0.156, $f^2$ = 0.03, 95% CI [0, 0.19]; **Appendix 1—figure 10C**).

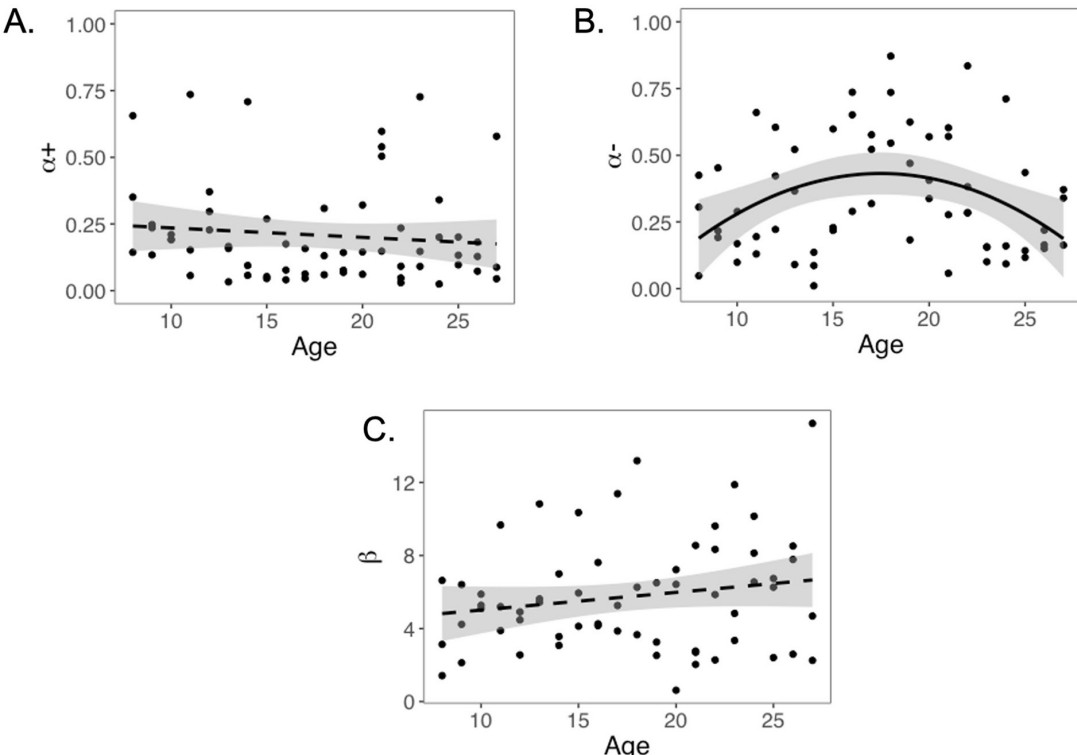

**Appendix 1—figure 10.** Age patterns in risk-sensitive temporal difference (RSTD) model parameters. Age-related change in (**A**) $\alpha^+$, (**B**) $\alpha^-$, and (**C**) $\beta$ parameter estimates from the RSTD model.

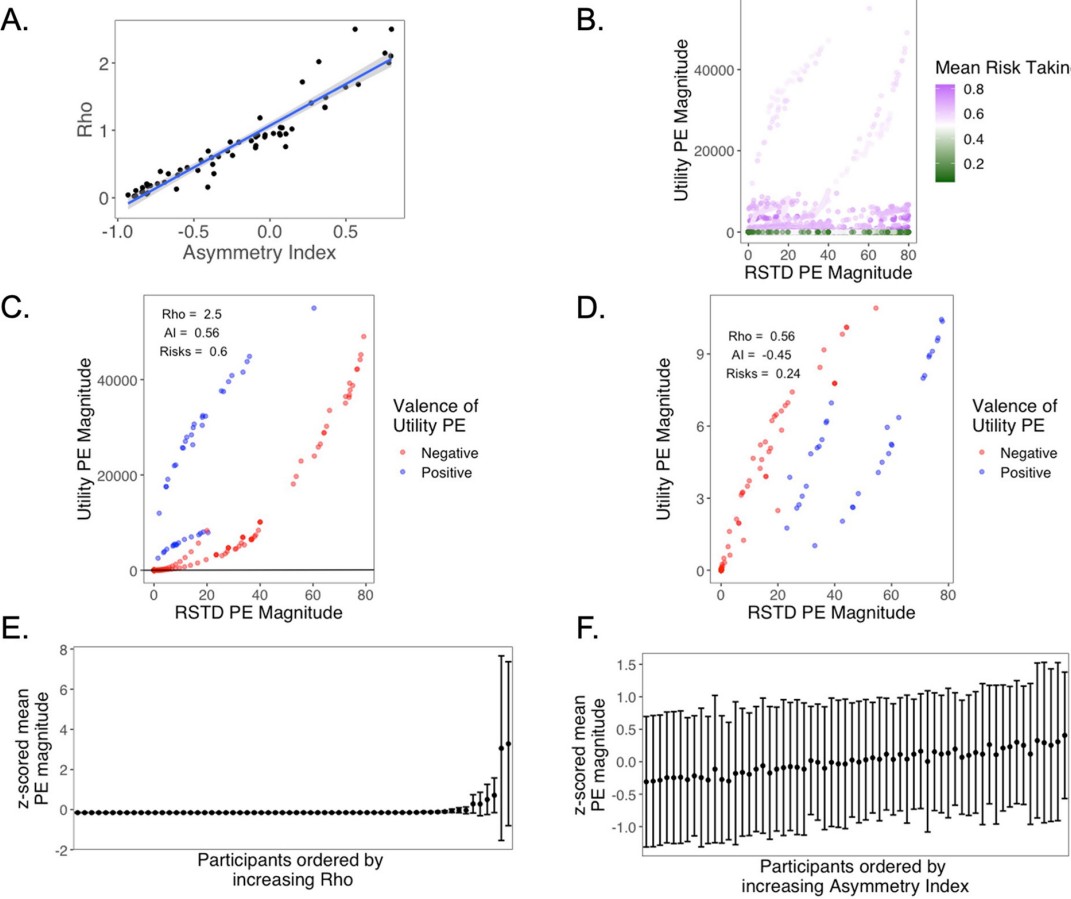

**Appendix 1—figure 11.** Relationship between parameter estimates and PEs derived from the risk-sensitive temporal difference (RSTD) and Utility models. (**A**) Relationship between asymmetry index (AI) and Rho. (**B**) PEs for all participants, colored by the mean proportion of risk taking in the task. Purple dots are PEs from risk-seeking participants, while green dots are from risk-averse participants. (**C**) PEs for an example risk-seeking participant. (**D**) PEs for an example risk-averse participant. (**E**) Mean PE magnitudes, z-scored within the full sample, and then averaged within-subject, ordered by increasing Rho. (**F**) Mean PE magnitudes, z-scored within the full sample and then averaged within-subject, ordered by increasing AI.

## Utility and subsequent memory

We examined whether PEs and choice biases indexed by the Utility model could explain the memory data similar to the observed relation between subsequent memory and PEs and valence biases derived from the RSTD model reported in the main text. To this end, we first assessed the relationship between RSTD and Utility model parameters. We found that AI, derived from the RSTD parameter estimates, and $\rho$ from the Utility model were highly correlated ($r = 0.95$, $p<0.001$; *Appendix 1—figure 11*), suggesting that they provided comparable individual difference measures.

However, due to the nonlinear transformation of outcome values within the Utility model, the two models yield very different value estimates and PEs. PE magnitudes derived from the Utility model also vary widely across participants. This pattern is clear in *Appendix 1—figure 11B*, in which we have plotted the PEs across all participants from both the RSTD and Utility models. The dot color represents proportion risk taking. The purple dots are PEs from risk-seeking participants, while green dots are from risk-averse participants, with lighter dots representing people who took risks on about half of trials and darker dots representing those whose risk taking deviated more from equal numbers of risky and safe choices. Because of the nonlinear Utility function, Utility PE magnitudes are much higher for risk-seeking participants (for whom $\rho > 1$) versus risk-

averse participants (for whom $\rho < 1$). In contrast, PE magnitudes from the linear RSTD model are necessarily constrained between 0 and 80 and are thus more uniformly distributed across participants regardless of risk preference. *Appendix 1—figure 11* displays the relationship between PEs from Utility and RSTD models from example risk-seeking (*Appendix 1—figure 11C*) and risk-averse (*Appendix 1—figure 11D*) participants. The participant represented in *Appendix 1—figure 11C* chose the risky option on 60% of trials in which the EVs of risky and safe options were equal. This participant's $\rho$ estimate from the Utility model was 2.5, and the RSTD AI was 0.56. In contrast, the participant in *Appendix 1—figure 11D* took risks on 24% of equal-EV risk trials and had a corresponding $\rho$ of 0.56 and AI of –0.45. The PEs from the two models are quite different in both absolute and relative scale. For the risk-seeking participant, the nonlinear transformation of outcome values means that Utility PEs reached magnitudes over 50,000, while RSTD PEs could only reach a maximum magnitude of 80.

We next tested whether the $\rho$ parameter and PEs derived from the Utility model could account for subsequent memory in the same manner that we observed for the RSTD model. As $\rho$ and AI are so highly correlated, any difference in explanatory ability of the two models would necessarily stem from differences in the ability of their distinct PE dynamics to capture trial-by-trial variability in subsequent memory. In contrast to the RSTD results, we did not find a significant three-way interaction between $\rho$, PE magnitude, and PE valence (using the PEs derived from the Utility model; $z = 1.06$, p=0.291, OR = 1.18, 95% CI [0.87, 1.60]).

This absence of an effect likely stems from the large variability in PE magnitude across participants within the Utility model (*Appendix 1—figure 11E*) than the RSTD model (*Appendix 1—figure 11F*). Under the Utility model, risk-seeking participants (high $\rho$) experience much larger magnitude PEs, while PEs for risk-averse participants are smaller in magnitude. Thus, within the multilevel model, high $\rho$ participants have not only greater means but also greater variance in their PEs, leading to an inherent prediction that risk-seeking participants should experience the strongest PE-dependent memory modulation. However, in our prior analysis using the RSTD model, valence biases in the effects of PE sign and magnitude on subsequent memory were evident in both risk-seeking and risk-averse participants. This suggests that only the linear values within the RSTD model can adequately capture this pattern.

## The effect of agency on learning

Prior studies have demonstrated that learning asymmetries can vary as a function of whether choices are free or forced, such that participants tend to exhibit a greater positive learning rate bias ($\alpha^+ > \alpha^-$) for free choices and either no bias or a negative bias ($\alpha^- < \alpha^-$) in forced choices (*Chambon et al., 2020*; *Cockburn et al., 2014*). To understand whether our participants exhibited different learning biases in free and forced trials, we implemented an additional reinforcement learning model with four learning rates (FourLR). Here, we used separate $\alpha^+$ and $\alpha^-$ for forced and free choices. Although this least-parsimonious model provided a better fit to the choice data than the TD model, it performed worse than both the Utility and RSTD models at both the group and participant level.

Free- and forced-choice learning rates and asymmetry indices are plotted in *Appendix 1—figure 12*. Consistent with prior findings (*Chambon et al., 2020*; *Cockburn et al., 2014*), we found that $\alpha^+$ was significantly higher in free (median = 0.22) compared to forced (median = 0.14) choices (Wilcoxon signed-rank test: $V = 1338$, p=0.011). In contrast, $\alpha^-$ was not significantly different in free (median = 0.35) vs. forced (median = 0.32) choices (Wilcoxon signed-rank test: $V = 794$, p=0.202). We computed separate AIs for free and forced choices, and found that AI was significantly higher for free (median = –0.12) compared to forced (median = –0.39) choices (Wilcoxon signed-rank test: $V = 1377$, p=0.005). Although $\alpha^+$ and AI were higher in free versus forced trials, median AIs were negative for both free and forced choices. Therefore, in this study, we did not observe positive learning rate asymmetries for free choices.

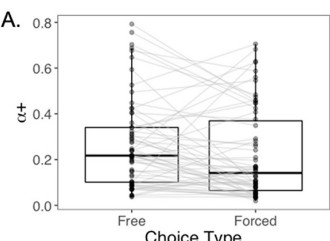 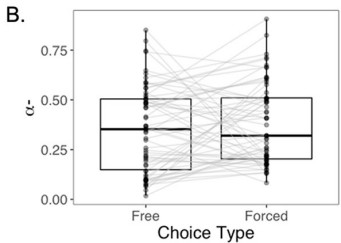 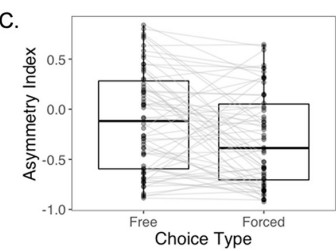

**Appendix 1—figure 12.** Learning parameters for free and forced choices. (**A**) Negative learning rates, (**B**) positive learning rates, and (**C**) asymmetry indices from the FourLR model are plotted as a function of choice type (free or forced).

## The effect of agency on memory

Prior research has also found better subsequent memory for memoranda associated with free versus forced choices (*Katzman and Hartley, 2020*; *Murty et al., 2015*). The present study was not designed to test the effects of agency on memory. The purpose of our forced-choice trials was to ensure all participants had similar opportunities to learn probabilities and outcomes associated with each probabilistic and deterministic point machine, regardless of their risk preferences. However, to test whether memory was different for items that appeared with outcomes of free versus forced choices, we ran two additional glmer models. First, we ran the most complex model included in the original manuscript (the model predicting memory accuracy that tested for a PE valence × PE magnitude × AI interaction, and also included linear and quadratic age and memory trial number), adding a predictor that indicating whether the image appeared after a free or forced choice. Memory did not vary as a function of choice agency ($z$ = –0.44, p=0.664, OR = 0.99, 95% CI: [0.95,1.04]).

As a final test for a potential agency benefit on memory, we tested whether the three-way PE Valence × PE Magnitude × AI interaction predicting memory performance varied as a function of whether the choice was forced or free (i.e., we tested for a four-way PE Valence × PE Magnitude × AI × choice agency interaction). The four-way interaction was also not significant (p=0.136). To further explore whether agency had any apparent qualitative effect, we plotted the (three-way interaction effect – PE Valence × PE Magnitude × AI) separately for free versus forced choice trials (*Appendix 1—figure 13*). Strikingly, we found that for free-choice trials (*Appendix 1—figure 13A*), which comprised the majority of trials, the interaction looks qualitatively similar to the overall three-way interaction effect across all trials that we report in the main text (*Figure 4B*, shown in *Appendix 1—figure 13C*). However, for forced trials (*Appendix 1—figure 13B*), memory performance for images that coincided with positive PEs did not appear to be modulated by AI. This pattern is similar to the memory effect observed in Experiment 2 (*Figure 6B*, shown in *Appendix 1—figure 13D*), in which participants provided explicit predictions of outcomes, but did not make choices. This qualitative similarity suggests that memory for surprising high-magnitude positive outcomes may be equivalently facilitated for all individuals in situations where they do *not* have agency, regardless of their idiosyncratic biases in learning.

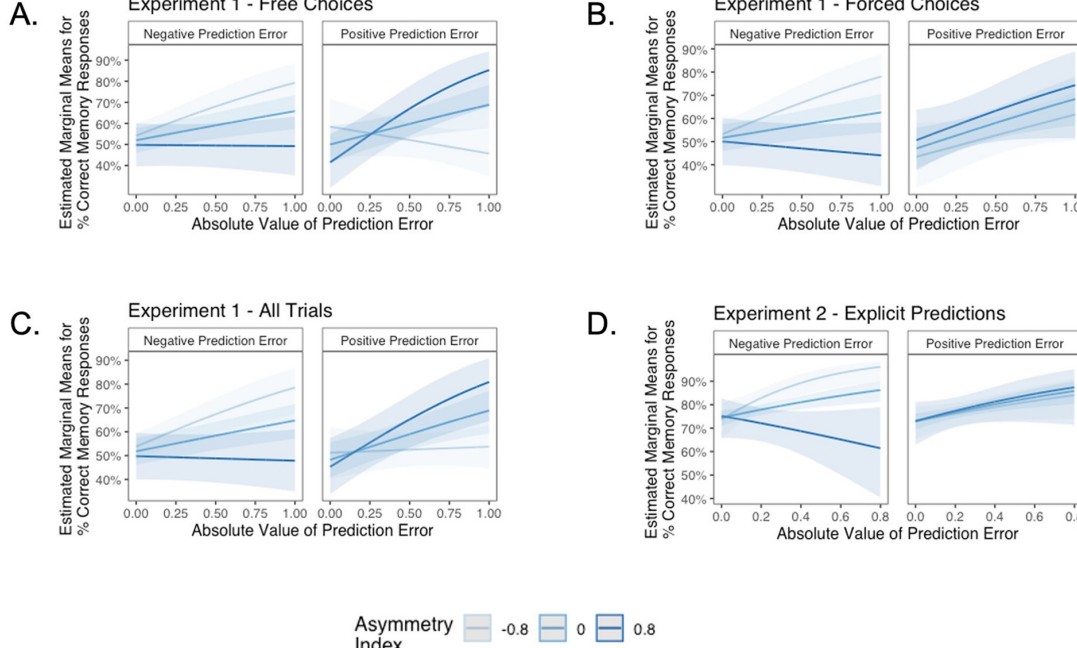

**Appendix 1—figure 13.** Learning biases and subsequent memory as a function of agency. (**A**) PE Valence × PE Magnitude × AI for free choices. (**B**) PE Valence × PE Magnitude × AI for forced trials. (**C**) PE Valence × PE Magnitude × AI for Experiment 1 (***Figure 4B***). (**D**) PE Valence × PE Magnitude × AI for Experiment 2 (***Figure 6B***). Note that the interaction effect for forced choices (**B**) resembles that in Experiment 2 (**D**) where participants were not asked to make choices.

## Experiment 2

To test for generalizability of our finding that learning biases affect what information is encoded in subsequent memory, we ran a modified version of our analysis from Experiment 1 on a previously published dataset (***Rouhani et al., 2018***). There were 383 adult participants in the dataset, each of whom completed one of the three experiments reported in the original manuscript.

### Pavlovian learning and memory task

In this study, participants completed a Pavlovian learning and memory task where they learned, through trial and error, about the average value of indoor and outdoor scene images. On each trial, participants viewed a scene image and provided an estimate for the average value of that image category, after which they were shown the true value of that image. After completing learning trials, participants completed a memory test for half of the scenes presented during learning and an equal number of new images.

There were three different versions of the task that varied in the number of trials and the variability of the outcome distributions. Each participant completed only one of the three versions. As the differences between the versions of the task were minimal, we modeled the data of participants of all three task variants as a single sample. A detailed explanation of the methods can be found in the original manuscript (***Rouhani et al., 2018***).

### Reinforcement learning model

We fit an Explicit Prediction temporal difference reinforcement learning model to the estimation data. This model was similar to our RSTD model from Experiment 1, in that it included separate learning rates for trials with positive ($\alpha^+$) and negative ($\alpha^-$) prediction errors with priors (Beta(1.2,1.2)) on both parameters. In this model, the value of each image category ($i$) on each trial ($t$) was estimated as $V_i(t + 1) = V_i(t) + \alpha^+ * \delta(t)$ when $\delta > 1$ and $V_i(t + 1) = V_i(t) + \alpha^- * \delta(t)$ when $\delta < 1$. Each participant's AI was calculated as in Experiment 1: AI = $(\alpha^+ - \alpha^-)/(\alpha^+ + \alpha^-)$.

## Model fitting

Participants' estimates and trial outcomes were rescaled between 0 and 1 prior to model fitting. Image category values were initialized at 0.5. We regressed model image value estimates on participants' actual estimates and minimized the negative log likelihood of that regression result for each participant using the optimization function fminunc in MATLAB.

## Assessment of model fit and exclusions for poor model fit

We computed estimated value on each trial using the best-fitting parameter. We computed the correlation between the model's value estimates and actual value estimates from participants' task experience as a metric of the quality of fit of the model to participants' data. The mean correlation was 0.42 (SD = 0.34). For 78 participants, there was either a negative correlation between model-derived and actual value estimates or the negative log likelihood of their data using the best-fit parameters was positive, or both, indicating that the model did not fit the data well. Because our analysis relies on trial-by-trial PEs estimated using best-fit model parameters, we removed participants from the dataset who were poorly fit by the model according to either model fit metric. However, we conducted a sensitivity analysis including all participants, below, to ensure that these exclusions did not drive any observed effects in the restricted sample. The mean correlation between estimated and actual PEs for the remaining 305 participants was 0.54 (SD = 0.22).

## Multilevel model fitting

The maximal model did not converge. As in Experiment 1, we reduced the model until we found a model that converged (*Barr et al., 2013*). The maximal model that converged is:

Memory Response~ Memory Trial Number + False Alarm Rate + AI * PE Valence * PE Magnitude + (1+ PE Valence + PE Magnitude + Memory Trial Number || SubjectNumber)

Below, we also report a sensitivity analysis that includes all participants, regardless of model fit. The model that converged when all participants were included is:

Memory Response~ Memory Trial Number + False Alarm Rate + AI * PE Valence * PE Magnitude + (1+ PE Magnitude + Memory Trial Number || SubjectNumber)

## Model parameters and asymmetry index

Distributions of learning-rate parameters from our model, as well as the distribution of our AI metric, are plotted in *Appendix 1—figure 14*. These distributions include all participants who were included in the analysis in the main text.

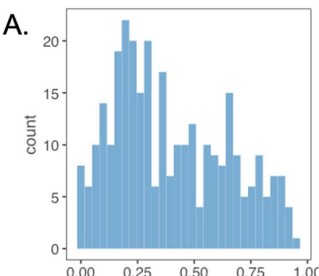 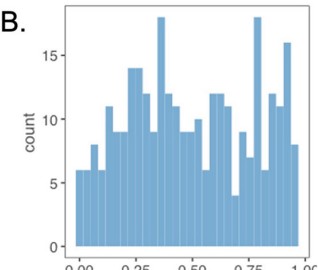 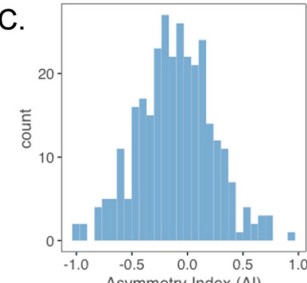

**Appendix 1—figure 14.** Distributions of parameters from the Explicit Prediction model. Distributions for (**A**) $\alpha^+$, (**B**) $\alpha^-$, and (**C**) asymmetry index (AI) in Experiment 2.

## Sensitivity analysis with all participants

To determine whether the observed relation between AI and memory was influenced by participant exclusions, we ran a sensitivity analysis fitting the multilevel model that included all 383 participants, including those poorly fit by our RL model (*Appendix 1—figure 15*). The three-way interaction between AI, PE valence, and PE magnitude predicting memory accuracy was marginally significant (z = 1.80, p=0.072, OR = 1.05, 95% CI [1.00, 1.11]; *Appendix 1—figure 15B*). In this model, there was also a significant two-way interaction between AI and PE valence (z = 2.83, p=0.005, OR = 1.08, 95% CI [1.02, 1.14]). Both the three-way and two-way interaction effects are

consistent with our hypothesis that individual's valence biases during learning (i.e., AI) modulate the relationship between PE valence and memory: those with negative learning biases had better memory for images that coincided with negative PEs and vice versa (*Appendix 1—figure 15C*). As in both Experiment 1 and in the analysis reported in the main text, there was a highly significant main effect of PE magnitude ($z = 5.30$, p<0.001, OR = 1.18, 95% CI [1.11, 1.26]).

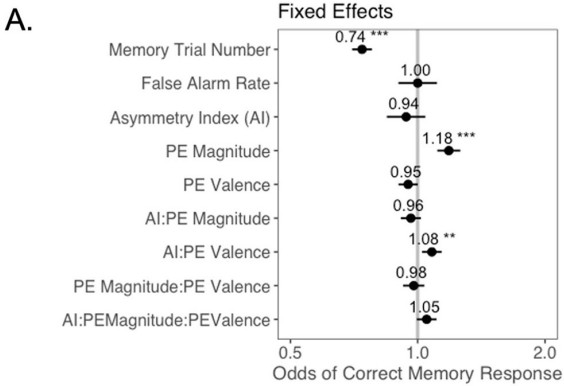

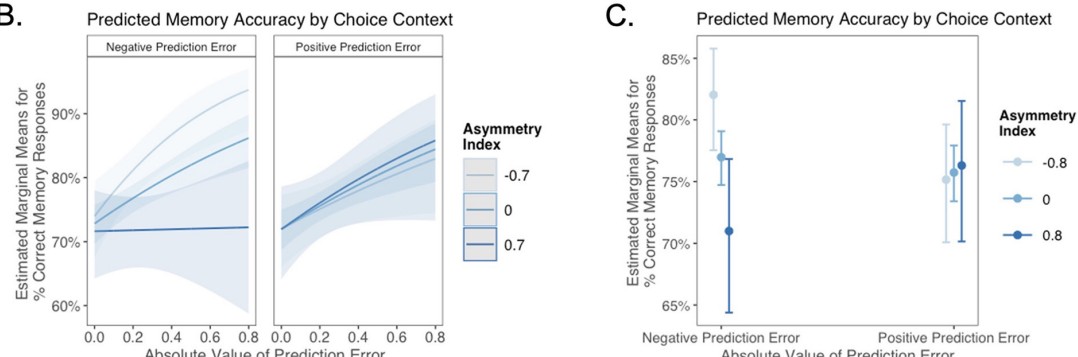

**Appendix 1—figure 15.** Experiment 2 sensitivity analysis. Generalized linear mixed-effects regression results demonstrating incidental memory accuracy for pictures presented during learning as a function of PE valence, PE magnitude, including participants poorly fit by the RL model. (**A**) Fixed-effects results. Whiskers represent 95% CI. (**B**) Estimated marginal means plot showing the marginally significant three-way interaction between asymmetry index (AI), PE valence, and PE magnitude. Shaded areas represent 95% CI for estimates. (**C**) Estimated marginal means for significant two-way interaction between AI and PE valence. Whiskers represent 95% CI. **p < .01, ***p < .001.

