## [Editor Report]

This paper will be of interest to cognitive and behavioral neuroscientists, behavioral economists, and developmental psychologists. The authors provide novel evidence that adolescents, relative to children and young adults, are prone to making risk-averse decisions because they are more attuned to negative outcomes during learning. The paper presents rigorous computational analyses that conclusively support the major claims and advance our understanding of age-related shifts in decision making.

---

## [Decision Letter]

**Decision letter after peer review:**

Thank you for submitting your article "Valence biases in reinforcement learning shift across adolescence and modulate subsequent memory" for consideration by *eLife*. Your article has been reviewed by 3 peer reviewers, one of whom is a member of our Board of Reviewing Editors, and the evaluation has been overseen by Michael Frank as the Senior Editor. The reviewers have opted to remain anonymous.

Essential revisions:

1. All three Reviewers had concerns about the modelling results. Many of these concerns stemmed from the lack of consideration of competing accounts for the data and/or limited justification provided for the choice of models. Please (a) clarify the use of the Pavlovian reinforcement learning model in Experiment 2 (Reviewer 1), (b) relate the current work to the utility model originally presented in the Niv 2012 paper which introduced this behavioural paradigm (Reviewer 3), and (c) add fits of additional competing models. Specifically, with regards to point (c), fitting the subjective value model to account for prospect theory, and the subjective utility model, would be informative.

2. Please account for the effect of forced vs. choice trials in both reinforcement learning and memory. All Reviewers highlighted this as an important consideration, as there is evidence that this distinction could lead to differential learning, attention, and/or memory (as shown previously by this group, e.g., Katzman & Hartley 2020).

3. Please include additional data visualizations that depict the raw data, which will better equip the reader for interpreting the results (e.g., the learning curves per experimental condition, and/or per age group as suggested by Reviewer 2).

4. Please consider the role of attention versus reward prediction error (RPE) in the memory effects. Reviewers pointed out that there may be attentional differences across probabilistic versus deterministic trials that could be driving the effects rather than RPE, which if the source of memory differences would warrant a different conclusion than the current paper. In addition, whether and how attention fits into the relationship between asymmetry index and memory was unclear. The authors may be in a position to address this question by performing additional analyses on the current data (e.g., by assessing whether RPE strength alone is related to encoding within the probabilistic trials, as suggested by Reviewer 2); or more generally (in the absence of data) clarify their position or speculate on these issues.

5. Please improve the integration of Experiment 2 into the paper. More details were needed for understanding the purpose of this sample. In addition, Reviewers noted that the interaction found in Experiment 1 did not entirely replicate in Experiment 2. This warrants more consideration in the paper.

6. Please clarify the task and methods descriptions. For example, it is not clear whether the forced choices were included in the modelling. Moreover, please ensure all details needed to understand the major points of the paper are included in the main text without requiring readers to reference the methods.

7. As highlighted by Reviewer 3, it is unclear whether asymmetry biases are a trait-stable characteristic, or rather would change with age within an individual. Please speak to this point as well as other limitations associated with the cross-sectional nature of this study in the revised Discussion.

8. The Reviewers had several additional thoughtful suggestions for other work the authors might cite. I recommend the authors consider whether these suggestions might enhance the novelty, impact, and/or clarity of their paper.

9. Please provide additional details on the models (e.g., reporting parameter β, age effects, model validation, and/or more information on the recoverability analysis) as appropriate. One specific question raised by reviewers was whether learning rate parameter estimation might be more difficult in participants with more deterministic choices and thus fewer trials with non-zero RPEs. Is this empirically true, i.e., does model parameter recovery (as shown in Table 1 overall) differ across ages/ranges of behavior?

*Reviewer #1:*

Rosenbaum et al. report an investigation of developmental differences in biases to learn from positive versus negative prediction errors-that is, experiences that are either better (positive) or worse (negative) than expected. One of their key findings is that adolescents show a bias towards negative outcomes in both reinforcement learning and memory, a finding that is somewhat surprising given teens' heightened sensitivity to reward and real-world risk-taking. That is, adolescents (1) showed a general bias to make riskier choices (selecting probabilistic rather than deterministic "machines" of equal average value) and (2) show greater learning from negative than positive prediction errors (i.e., showed a negative asymmetry index [AI]). In addition, individual differences in AI were mirrored by a similar memory advantage, where participants showing greater positive AI likewise showed a bias to remember more surprisingly positive stimuli; whereas those showing a greater negative AI were biased to remember more surprisingly negative stimuli. These individual differences were unrelated to age.

This is a strong paper with an interesting set of results. The authors additionally partially replicated their individual difference findings in a separate dataset (experiment 2; including only adults but a larger sample than the main experiment 1), which is a notable strength and somewhat increases confidence in the findings. I am particularly struck by the surprising finding that adolescents are biased towards negative rather than positive outcomes. As appropriately noted in the paper-and particularly given the adolescents did self-report a relatively higher amount of real-world risk taking than other ages, suggesting that it is not the case that the teens in the study are atypically risk-averse in their everyday lives-the degree to which the reinforcement learning task accurately captures something akin to everyday risk taking warrants further consideration in the field.

In addition to these strengths, the paper has several weaknesses as follows:

1. I am left unsure as to the value added by the reinforcement learning model and whether the asymmetry index reflects something unique about learning as compared with a general bias towards deterministic choices in the adolescent period. That is, if the authors wish to make claims about asymmetry of learning rate specifically (α) it would be important to know if this is dissociable from the choice or sampling bias they observe. Empirically, is it the case that there would be a similar set of findings if one focused on the percent probabilistic choices value per participant (data in Figure 2) rather than the asymmetry index (Figure 3)? If these are somewhat redundant or overlapping metrics, I would encourage the authors to clarify this in the paper and perhaps only focus on one of them in the main text, so as not to imply these are separate findings.

2. Related to my above point, those individuals who make fewer probabilistic choices (disproportionately adolescents) have fewer opportunities to learn from prediction errors that are either positive or negative. That is, in my understanding, there will be no prediction error from the deterministic machines which the adolescents primarily select. Their bias to select deterministic machines seems fairly extreme; it appears as though they shy away from the probabilistic choices across the board, even when the value of the probabilistic choice on average would greatly exceed that of the deterministic choice (e.g., as shown in Figure S2). As such, I am wondering whether theoretically the authors can clarify the relationship between choice and learning rate; and analytically whether the bias towards deterministic choices could impact the model fitting procedures.

3. As an additional interpretive question, I had trouble linking the asymmetry index to the memory performance and understanding how the authors believed these metrics to be related. Is there a general increase in interest or attention or salience when a prediction error aligning with the learner's biases (e.g., a highly positive prediction error if I'm someone who learns best from those types of trials – that is, showing a positive AI) happens, therefore indirectly impacting memory formation in the process? Or, is this thought to be a separate mechanism that occurs (the learning from positive prediction errors vs. "prioritization" in memory of those positively valenced experiences)? It seems to me as though both are related to learning/encoding, and thus this individual difference may not be terribly surprising, but I was unsure about the authors' preferred interpretation.

4. While I appreciated the inclusion of experiment 2, I felt that it was not particularly well integrated into the paper. For example, why did the authors opt to use data with only adults rather than a developmental sample (and does this constrain interpretation in any way)? In addition, it is important to highlight that the results do not fully replicate the main findings. In particular, there is no difference in the relationship between the magnitude of prediction error and memory among positively valenced trials according to AI, which was observed in the main developmental experiment 1. This discrepancy warrants more attention in the paper. (Could this be about the sample characteristics? Task differences? and so on.)

5. It was not clear at the conceptual level how the Pavlovian reinforcement learning model fit in experiment 2 is different from the main RSTD used in experiment 1, and/or why these paradigms required the use of different models. Additional description on this point would be helpful.

6. I would appreciate more context for the recoverability results. In particular, the ability to recover which model generated the data for simulated participants (RSTD vs. TD) seemed fairly low. I understand the point about low-asymmetry participants generated by the RSTD model being more parsimoniously fit by the simpler TD model, so that direction of error does not bother me so much. However, I am puzzled by the 87% correct for those simulated participants generated by the TD model. This would have to mean, I think, that the simple TD behavior was being incorrectly attributed to the more complex two-alpha model? I was hoping the authors could provide additional context or reassurance that this qualifies as "good" performance.

7. There were some details not sufficiently described in the main paper for the reader to understand without referencing the methods or supplement. For example: How did the forced versus choice trials work? What is "test trial performance" – the test trials are not described until the supplement and it was confusing to me because the only "test" mentioned in the main paper at this point was the memory test. How were the probabilistic vs. deterministic choices incorporated into the mixed-effects modeling?

8. How were the different responses on the memory test considered? There were four choices according to the figure but it is described as though it is simply "old" vs. "new" responses. Please clarify how this was handled in analysis as well as the reason for inclusion of these four options.

9. I apologized if I missed this point in the paper: I understand that the models were fit to individual data and most participants were better fit by RSTD than TD. However, the authors also discuss the RSTD model winning overall and all subsequent analyses on the individual differences require the two separate α values. So, I am confused as to whether (a) all participants were ultimately fit with the RSTD so all could be included in the analysis, despite some being better fit by TD or (b) only participants who were best fit by RSTD were included in subsequent analyses (or of course (c) something else)? I think this would be OK because my assumption would be that the participants would simply have two α values (positive and negative) that would be similar if their behavior is better explained by the TD model, but I just wanted to clarify.

*Reviewer #2:*

The authors investigate valence biases in reinforcement learning (RL) and memory, and how they change across adolescence. In a medium size developmental study where outcomes are learned rather than instructed (n = 62, ages 8-27), the results show a surprising choice pattern: adolescents have a stronger bias to select a choice with a deterministic 40 outcome, over a probabilistic 80/0 (50%), compared to younger and older participants. The authors interpret this as a more risk-averse performance, and operationalize it as a difference in RL learning rate. Memory test phase results show that individual images observed with a high absolute reward prediction error (RPE) are more likely to be recalled, and that individuals' valence biases predict a valence asymmetry of this RPE effect. The latter result is partially replicated in a reanalysis of a previously published, non-developmental study.

This is a well-written, easy to read article, that adds a new data point to the complex and contradictory literature on valence, risk, and adolescence, without resolving it. The strengths of the article are (1) its interesting experimental protocol and pure behavioral results (2) the existence of a replication of the complex memory finding.

There are also a number of weaknesses.

First the behavioral results are presented in a very processed manner, limiting the ability to re-interpret the results.

Second, the computational modeling is fairly limited and lacking in competing accounts, which also limits the interpretation of the results. As such, it seems that at this point, the results do not fully support the conclusions – in particular, it is not clear that the interpretation in terms of asymmetric learning rates is warranted yet.

Comments for the authors:

1. Presentation of the results. All figures presented are quite removed from the raw data. It would be helpful to provide more unprocessed results – for example, the learning curves per experimental condition, and/or per age group. This would also provide a more sensitive target for model validation than the ones currently presented in Figure S4. It is much harder for the reader to interpret the results when only very processed data is shown.

This is a well-written, easy to read article, that adds a new data point to the complex and contradictory literature on valence, risk, and adolescence, without resolving it. I see a number of issues with the presentation of the results, the modeling and interpretation of the results.

2. Modeling. The authors use two very simple RL models to capture the performance (a classic δ rule model, and the same with two learning rates). There are a few relevant aspects to the modeling that are either not considered or not reported, but that are important for interpreting the results.

a. Please indicate Q-value initialization and reward rescaling as part of the model description, rather than as part of the model fitting procedure. The initialization has theoretical impacts, so should be part of the model.

b. Prospect theory indicates that a value of 80 is subjectively worth less that 2* the subjective value of 40. As such, the claim that "expected value" and "risk" are controlled for is debatable: if a true 40 feels like a 50 in comparison to an 80, then the two "same objective expected value stimuli" have different subjective expected values, without a need to invoke risk. The authors should test a competing model where the learning rate is fixed, but the obtained reward is modified according to prospect theory (i.e. subjective reward = 80*((reward/80)^p), where p is a free parameter). This model should also be able to capture the presented processed results (existence of an apparent "risk-aversion"), but should have slightly different temporal dynamics, and would lead to a completely different interpretation of the results.

c. Please report parameter β, age effects. Also provide more model validation.

d. Have the author investigated the effect of forced vs. free choice trials, both on RL and memory? There is evidence that this leads to differential learning processes, and potentially differential attention, which could impact both the learning and memory findings.

3. Memory findings:

a. Can the authors comment on the role of attention? Presumably, the participants pay more attention to the outcome of probabilistic than deterministic choices. Could this be a factor in encoding the image, instead of the actual RPE strength? Is there some analysis in probabilistic trials that could be done to show convincingly that the actual size of the RPE matters? In the current analysis, it seems confounded with condition.

b. While the overall statistical pattern is replicated (Figure 4 and 6A), the realization of the triple interaction looks different in the two experiments (Figure 4 and 6B). In the replication, the asymmetry seems to not matter for positive RPEs, and to have a stronger effect for negative RPEs. For the new study, the patterns seem symmetrical between positive and negative RPEs. Can the authors comment on this?

*Reviewer #3:*

This work provides novel insights about how good and bad feedback differentially influence risky decision making with age from childhood to young adulthood. Further the authors examine how valenced learning biases (updating from positive or negative feedback) influence subsequent memory. The authors tested individuals age 8 to 27 using a risk-sensitive learning task. The task design allowed for the authors to measure learning asymmetries following choices with equivalent expected value but different levels of risk. Learning feedback was accompanied by trial unique images, which allowed for an examination of how learning biases modulate memory. They found that adolescents made more risk-averse decisions, and they over-weighted negative prediction errors during learning. Memory analyses revealed that individual differences in asymmetry bias influenced subsequent memory, an effect that was replicated in an independent sample. The results strongly support the conclusions put forth in this paper. These findings provide timely contributions to both developmental cognitive neuroscience and contemporary research in memory and decision making. The current work provides important revisions to theoretical models in both fields.

Strengths

Prevailing theories of cognitive development posit that adolescents are uniquely attuned to rewards. Yet, little research has investigated developmental differences in sensitivity to negative feedback. The present study advances our understanding of how valence asymmetries (learning from positive vs. negative feedback) during learning bias decision making and subsequent memory across development. The authors use an experimental paradigm that assesses risk-sensitive reinforcement learning. This approach is well-suited to identify individual biases in using positive and negative feedback to guide decision making. The results provide novel, yet surprising, evidence that adolescents are more risk-averse due to enhanced sensitivity to negative feedback. These findings offer an important revision to the current theoretical models of adolescent behavior.

The subsequent memory findings advance our understanding of how prediction errors during learning modulate memory encoding. Prior work has produced mixed results as to whether the valence or magnitude of prediction error influences memory. Here, the authors illustrate that individual differences in learning biases can account for when outcome valence modulates memory. The authors report that asymmetric biases in updating from positive vs. negative prediction errors during learning modulate subsequent memory. However, this effect depends on individual differences in learning asymmetries. Therefore, reward enhances subsequent memory for individuals who are more sensitive to positive prediction errors, however punishment enhances subsequent memory in individuals who are more sensitive to negative prediction errors. A major strength of this paper is that the authors reproduce this memory finding by conducting novel analyses in an existing dataset that was collected in an adult sample.

A major strength of this paper is the analytical approach. The authors implement rigorous computational modeling procedures. They test multiple models of learning and run formal model comparisons to identify the best fitting model. The authors also run simulations and present parameter and model recovery. To test age-related differences, the authors fit both linear and nonlinear age terms. Moreover, to confirm the strength of nonlinear effects, they conducted break-point analyses using segmented regression, as recommended by the Simohnson two-line approach.

Weaknesses

This paper presents data from a cross-sectional sample. This raises questions as to whether learning asymmetries are a stable individual characteristic, or whether these biases exhibit within-person changes with age. Nonetheless, the results of this paper provide important advances to our understanding of age-related differences in learning, decision making, and memory that can form the basis of future longitudinal studies.

Comments for the authors:

This manuscript is exceptionally well-written, and the authors present the methods and findings very clearly. I commend the authors approach to computational modeling and nonlinear age analyses. The present findings provide exciting and novel insights about how adolescents approach risky decision making, which in turn has consequences for memory formation. This is a strong paper, and I believe that the current conclusions warrant publication in *eLife*. However, I also believe that the inclusion of some additional analyses and clarifications, which I offer below, will further strengthen this manuscript.

In the introduction, the authors explain how prior research in developmental samples cannot disentangle learning asymmetries because performance in prior tasks improved if individuals relied upon updating from positive prediction errors. To clarify this point, and to emphasize the novelty of the current study, it would be helpful if the authors provide a more detailed explanation as to how the present design differs from the paradigm used in Van den Bos 2012.

In the present paper, the authors run model comparison for a basic TD model and a risk-sensitive reinforcement learning model. However, risk sensitivity may be influenced by nonlinear utility functions. It would be informative if the authors also discussed the utility model, as presented in the Niv 2012 paper, which first introduced the present behavioral paradigm. If the authors did not fit this model, please provide an explanation as to why this model was not tested in the current sample.

Prior work from this group has shown that choice and agency can influence memory (e.g. Katzman & Hartley 2020). Therefore, for the memory data, it would be helpful if the authors accounted for the different trial types (choice trials vs. forced trials) in the analyses.

Due to the cross-sectional nature of the sample, it is unclear if asymmetry biases are a trait-stable characteristic, or whether this bias changes with age within an individual. It would be helpful for the authors to address this in the discussion.

---

## [Author Response]

Essential revisions:1. All three Reviewers had concerns about the modelling results. Many of these concerns stemmed from the lack of consideration of competing accounts for the data and/or limited justification provided for the choice of models. Please (a) clarify the use of the Pavlovian reinforcement learning model in Experiment 2 (Reviewer 1), (b) relate the current work to the utility model originally presented in the Niv 2012 paper which introduced this behavioural paradigm (Reviewer 3), and (c) add fits of additional competing models. Specifically, with regards to point (c), fitting the subjective value model to account for prospect theory, and the subjective utility model, would be informative.

We thank the reviewers for these helpful suggestions to clarify our modeling results. Below we address each of the three points (a-c) individually in turn.

a) A critical distinction between Experiments 1 and 2 is that, while participants in Experiment 1 made choices between point machines that influenced the outcomes they observed, participants in Experiment 2 provided explicit numerical predictions of the likely outcome value for each stimulus image, and the observed outcomes were drawn from a fixed distribution. We referred to the Experiment 2 model as “Pavlovian” because participants made no choices, and thus our model was fit to participants’ trial-by-trial predictions rather than to choices. However, we now recognize that the terminology was confusing and we now refer to the model as the “Explicit Prediction” model.

On the implementation level, the RSTD model and the Explicit Prediction model differ in that the RSTD model requires an extra parameter, *β*. This extra parameter allows us to use the softmax function to convert the relative estimated values of the two machines into a probability of choosing each machine presented in Experiment 1, which is then compared to participants’ actual choices during maximum likelihood estimation. In contrast, in Experiment 2 participants explicitly report their value predictions (and do not make choices), so the model’s free parameters can be fit by minimizing the difference between the model’s value estimates and participants’ explicit predictions.

To clarify these differences, in the manuscript, we have changed all references to the “Pavlovian model” to the “Explicit Prediction model”, and we have augmented our description of the distinction between the Explicit Prediction and RSTD models as follows:

Results:

“To quantify valence biases in this task, we fit an “Explicit Prediction” RL model that was similar to the RSTD model used in Experiment 1, but was fit to participants’ trial-by-trial predictions rather than to choices. […] Results are reported in Figure 6.”

Methods:

“In order to derive each participant’s AI, we fit an “Explicit Prediction” RL model to the participants’ estimation data (see Appendix 1 for more details on our model specification and fitting procedure). […] In contrast, in Experiment 2, participants explicitly reported their value predictions (and did not make choices), so the model’s free parameters were fit by minimizing the difference between the model’s value estimates and participants’ explicit predictions.”

b) As noted by the reviewers, in the Niv et al., 2012 paper that informed the current study, an additional “Utility” model that converted outcome values into subjective utilities was also fit to the data. We now include a Utility model in the manuscript. This model is necessarily different from the Utility model presented in Niv et al. (2012) due to a critical difference between our experimental paradigms. Specifically, in Niv et al. (2012), there was only one pair of equal-EV risky/safe machines (safe 20 points or 50% 0 points, 50% 40 points). With this version of the paradigm, the authors assumed that “*U*(0)=0, *U*(20)=20, and *U*(40)=*a**20,” with smaller values of their utility parameter, *a,* consistent with a concave utility curve, and larger values of *a* consistent with a convex utility curve.

Our version of the task introduced a second pair of equal-EV risky and safe machines (safe 40 points or 50% 0 points, 50% 80 points). Because subjective utilities are nonlinear transformations of value, we could not apply the Utility model from *Niv* et al. (2012) to our present paradigm. Instead, we fit a model in which all outcomes were first transformed into utilities with an exponential subjective utility function (*U*(value)=value*^ρ^*; Pratt 1964) within the same value update equation used in the TD model.

We conducted simulations with the Utility model to assess parameter and model recoverability. As with the RSTD model, we simulated 10,000 participants with *α*, *ρ* and *β* drawn randomly from the distribution of fit parameters observed in our empirical data. We found that all parameters were comparably recoverable to those from the RSTD model (Utility: *α*: *r* = .75, *ρ*: *r* = .88, *β*: *r* = .88; RSTD: *α^+^*: *r* = .79, *α^-^*: *r* = .88, *β*: *r* = .90). Additionally, model recovery analyses indicated that the RSTD and Utility models were reasonably identifiable. 76% of simulated participants’ data generated by the Utility model were best fit by the Utility model, and 65% of simulated participants’ data generated by the RSTD model were best fit by the RSTD model.

Assessment of which model provided the best fit to participants’ data was inconclusive. Author response image 1 displays BICs for each participant for the RSTD and Utility models. At the group level, the *median* BIC for the Utility model was numerically lower than the RSTD model (Utility: 131.06, RSTD: 131.93). At the single participant level, 36 participants were best fit by the Utility model whereas 26 participants were best fit by the RSTD model. The median within-subjects ΔBIC between the two models was 0.33 while the mean ΔBIC was 1.30, both less than the metric of ΔBIC > 6 that is taken as evidence of a superior fit (Raftery 1995). Taken together, relative BIC metrics at the group and subject level did not provide strong evidence (i.e., ΔBIC > 6) in favor of either model.

**Author response image 1. sa2fig1:** BICs for each participant from RSTD and Utility models.

To examine whether one model provided a better qualitative fit to the data, we next conducted a posterior predictive check. Here, we tested the extent to which the choice predictions made by each model correlated with the actual choice behavior of participants. We simulated 100 subjects for every real participant’s empirically derived parameters, using both the Utility model and the RSTD model, and took the mean proportion of risks across all 100 simulated subjects for each model. Choices derived from RSTD model simulations using each participants best-fit parameter estimates exhibited a significantly stronger correlation with actual choices (*r* = .92) than those simulated using the Utility model (*r* = .89; *t*(61) = 2.58, *p* <.012). Thus, it appears that the RSTD model provides a better qualitative fit to the choice data compared to the Utility model.

We next asked whether one model might provide a better account of the observed relation between reinforcement learning biases and subsequent memory reported in our manuscript. We previously found that subsequent memory for images encountered during learning related to individual valence biases in learning. That is, we observed a significant 3-way interaction effect of PE valence, PE magnitude and the asymmetry index (AI; or the relative size of *α^+^* and *α^-^* derived from the RSTD model) on memory accuracy.

In order to examine whether choice biases indexed by the Utility model could similarly explain the memory data, we first assessed the relationship between RSTD and Utility model parameters. We found that the AI, derived from the RSTD parameter estimates, and *ρ* from the Utility model were highly correlated (*r* = .95, *p* <.001; Appendix 1—figure 11A), suggesting that they provided comparable individual difference measures.

However, due to the nonlinear transformation of outcome values within the Utility model, the two models yield very different value estimates and Pes. PE magnitudes derived from the Utility model also vary widely across participants. This pattern is clear in Appendix 1—figure 11B, in which we have plotted the PEs across all participants from both the RSTD and Utility models. The dot color represents proportion risk taking. The purple dots are PEs from risk-seeking participants, while green dots are from risk-averse participants, with lighter dots representing people who took risks on about half of trials and darker dots representing those whose risk taking deviated more from equal numbers of risky and safe choices. Because of the nonlinear Utility function, Utility PE magnitudes are much higher for risk-seeking participants (for whom *ρ* > 1) versus risk-averse participants (for whom *ρ* < 1). In contrast, PE magnitudes from the linear RSTD model are necessarily constrained between 0 and 80 and are thus more uniformly distributed across participants regardless of risk preference. Appendix 1—figure 11C and D display the relationship between PEs from Utility and RSTD models from example risk-seeking (Appendix 1—figure 11C) and risk-averse (Appendix 1—figure 11D) participants. The participant represented in Appendix 1—figure 11C chose the risky option on 60% of trials in which the EV of risky and safe options were equal. This participant’s *ρ* estimate from the Utility model was 2.5, and the RSTD AI was.56. In contrast the participant in Appendix 1—figure 11D took risks on 24% of equal-EV risk trials and had a corresponding *ρ* of.56 and AI of -.45. The PEs from the two models are quite different in both absolute and relative scale. For the risk-seeking participant, the nonlinear transformation of outcome values means that Utility PEs reached magnitudes over 50,000, while RSTD PEs could only reach a maximum magnitude of 80.

We next tested whether the *ρ* parameter and PEs derived from the Utility model could account for subsequent memory in the same manner that we observed for the RSTD model. As *ρ* and AI are so highly correlated, any difference in explanatory ability of the two models would necessarily stem from differences in the ability of their distinct PE dynamics to capture trial-by-trial variability in subsequent memory. In contrast to the RSTD results, we did not find a significant 3-way interaction between *ρ*, PE magnitude and PE valence (using the PEs derived from the Utility model *z =* 1.06, *p* = .291, *OR* = 1.18, 95% CI [0.87, 1.60]).

This absence of an effect likely stems from the large variability in PE magnitude across participants within the Utility model (Appendix 1—figure 11D) than the RSTD model (Appendix 1—figure 11E). Under the Utility model, risk-seeking participants (high *ρ)* experience much larger magnitude PEs, while PEs for risk-averse participants are smaller in magnitude. Thus, within the multilevel model, high *ρ* participants have not only greater means but also greater variance in their PEs, leading to an inherent prediction that risk-seeking participants should experience the strongest PE-dependent memory modulation. However, in our prior analysis using the RSTD model, valence biases in the effects of PE sign and magnitude on subsequent memory were evident in both risk-seeking and risk-averse participants. This suggests that only the linear outcome values within the RSTD model can adequately capture this pattern.

To summarize, although we found that the Utility and RSTD models provided a similar degree of fit to the learning data based on BIC, the difference in fit between the models was equivocal (median ΔBIC = .33, mean ΔBIC = 1.30; both less than ΔBIC of 6; Raftery 1995). Moreover, the RSTD model provided a better fit to the learning data (based on our posterior predictive check) and was uniquely able to account for individual differences in the memory data (based on our multilevel model results). Given the better predictive fit of the RSTD model to both the learning and memory data, we now present the comparison of both models within the revised manuscript, but we continue to focus the framing of the study and our central results in the main text on the RSTD model. However, we have also expanded the part of the discussion in which we discuss the relation between the conceptual constructs of asymmetric learning rates and nonlinear utilities as accounts of risk preferences.

In the main text Methods, we introduce the Utility model as an alternative model fit to the data and present the model comparison results. The details of the posterior predictive check, and model and parameter recoverability have been added to the supplement. We have also expanded our discussion of the conceptual relations between the dual learning rate and subjective utility frameworks.

Methods:

“Utility Model.

As a further point of comparison with the TD, RSTD, and FourLR models, we estimated a utility model that employed the same value update equation as the TD model, *Q_M_*(*t*+1) = *Q_M_*(*t*) + *α* * δ(*t*). […] In contrast, ρ>1 corresponds to a convex utility function that yields risk-seeking behavior.

Methods:

“Parameter and Model Recovery

For each model, we simulated data for 10,000 subjects with values of each parameter drawn randomly and uniformly from the range of possible parameter values. […] We also found that RSTD model parameters were reasonably well recovered across the range of AI observed in our empirical sample (see Appendix 1, Appendix 1—figure 8).”

Results:

“To better understand the learning processes underlying individuals’ decision making, we compared the fit of four Reinforcement Learning (RL) models to participants’ choice behavior. […] Because the RSTD model fit choice data approximately as well as the Utility model, provided a significantly better qualitative fit to the choice data, and yielded an index of valence biases in learning, we focused our remaining analyses on the RSTD model (see Appendix 1 for additional model comparison analyses, and for an examination of the relation between the Utility model and subsequent memory data).”

Discussion:

“Traditional behavioral economic models of choice suggest that risk preferences stem from a nonlinear transformation of objective value into subjective utility (Bernoulli, 1954; Kahneman & Tversky, 1979), with decreases in the marginal utility produced by each unit of objective value (i.e., a concave utility curve) producing risk aversion. […] However, a potential parsimonious account is that a risk-sensitive learning algorithm could represent a biologically plausible process for the construction of risk preferences (Dabney et al., 2020), in which distortions of value are produced through differential subjective weighting of good and bad choice outcomes (Mihatsch & Neuneier, 2002; Niv et al., 2012).”

Appendix 1:

“We ran a posterior predictive check on both the RSTD and Utility models. We simulated 100 subjects for every real participant’s empirically derived parameters, using both the Utility model and the RSTD model, and took the mean proportion of risks across all 100 simulated subjects for each model. […] Thus, it appears that the RSTD model provides a better qualitative fit to the choice data compared to the Utility model.”

Additionally, we included Appendix 1—figures 11A-F, along with the explanation preceding those figures in Appendix 1.

c) In addition to the TD, RSTD, and Utility models, we compared one additional model in response to Essential Revision 2 below concerning potential differences in learning from forced versus free choices. This model included Four learning rates (henceforth FourLR): *α^+^* and *α^-^* for free choices and *α^+^* and *α^-^* for forced choices. Although the FourLR model performed better than the TD model (see (p. 31, lines 1230-1232, Appendix 1—figure 5).; FourLR Median BIC: 141.25; TD Median BIC: 145.35), FourLR median BIC was worse than both RSTD and Utility models (Utility: 131.06, RSTD: 131.93). On the individual participant level the RSTD provided a better fit than FourLR for 57 out of 62 participants (Median ΔBIC = 8.59). The Utility model fit better than the FourLR model for 55 out of 62 participants (Median ΔBIC = 8.41). Moreover, although parameter recovery was reasonable for FourLR (Table 1), model recovery was poor (Table 2). That is, when we simulated participants with the FourLR model and fit the data using the alternative models, FourLR fit better than RSTD for only 31% of participants and Utility for only 39% of participants, suggesting that our experimental paradigm was not able to reliably distinguish the value computation process formalized within the FourLR model from those of these more parsimonious models. Since the RSTD and Utility models both performed better than the FourLR model in model comparison, we focus on RSTD and Utility in the main text. However, we added a brief description of the FourLR model to the main text and included additional analyses of FourLR parameters in Appendix 1 (see response to Essential Revision 2 for more information).

Finally, the suggestion was raised that we should fit both a Prospect Theory model and a subjective utility model. However, given the set of choices participants faced in our task, these two models would have equivalent explanatory power. All outcomes in our task were in the gain domain, so a loss aversion parameter could not be estimated. Moreover, we did not vary outcome probabilities of the stochastic choices in the task, and thus nonlinear probability weighting also could not be assessed. In the Utility model, the exponential value transformation by *ρ* is no different from Prospect Theory’s exponential value transformation in the gain domain. Given this formal equivalence in the context of our task, we did not compare any additional Prospect Theory model.

2. Please account for the effect of forced vs. choice trials in both reinforcement learning and memory. All Reviewers highlighted this as an important consideration, as there is evidence that this distinction could lead to differential learning, attention, and/or memory (as shown previously by this group, e.g., Katzman & Hartley 2020).

As noted by the reviewers, past studies (including work from our group) have demonstrated differential learning, attention, and memory when making free versus forced choices (Katzman and Hartley 2020; Cockburn et al. 2014; Chambon et al. 2020). Below we address the additional analyses and corresponding changes to the manuscript that we have made to address the myriad potential ways in which making free versus forced choices may have influenced our results.

Prior studies have demonstrated that learning asymmetries can vary as a function of whether choices are free or forced, such that participants tend to exhibit a greater positive learning rate bias (*α^+^* > *α^-^*) for free choices and either no bias or a negative bias (*α^+^*< *α^-^*) in forced choices (Cockburn et al. 2014; Chambon et al. 2020). To understand whether our participants exhibited different learning biases in free and forced trials, we implemented an additional reinforcement learning model with four learning rates (FourLR, briefly referenced above in response to Essential Revision 1C). Here, we used separate *α*^+^ and *α*^-^ for forced and free choices. Although this least-parsimonious model provided a better fit to the choice data than the TD model, the FourLR model performed worse than both the Utility and RSTD models at both the group and participant level (see Appendix 1—figure 5 in response to Essential Revision 1C) and it performed poorly in model recovery analyses.

Despite the model’s poor performance at fitting choice data relative to more parsimonious models, parameter recovery from the model was reasonable. Thus, we conducted an exploratory analysis of whether learning rates differed as a function of choice agency. Free- and forced-choice learning rates and asymmetry indices are plotted in Appendix 1—figure 12*.* Consistent with prior findings (Cockburn et al. 2014; Chambon et al. 2020), we found that *α^+^* was significantly higher in free (Median = .22) compared to forced (Median = .14) choices (Wilcoxon Signed-rank test: *V* = 1338, *p* = .011). In contrast, *α^-^* was not significantly different in free (Median = .35) vs. forced (Median = .32) choices (Wilcoxon Signed-rank test: *V* = 794, *p* = .202). We computed separate AIs for free and forced choices, and found AI was significantly higher for free (Median = -.12) compared to forced (Median = -.39) choices (Wilcoxon Signed-rank test: *V* = 1377, *p* = .005). Although *α^+^* and AI were higher in free versus forced trials, median AIs were negative for both free and forced choices. Therefore, in this study, we did not observe positive learning rate asymmetries for free choices.

In the main text, we now briefly mention the FourLR model in the Results (please see revisions in Essential Revision 1B in addition to revisions below), and Methods sections. These changes are detailed here:

Results:

“Prior work has found positive valence biases tend to be positive in free choices, but neutral or negative in forced choices (Chambon et al. 2020; Cockburn et al. 2014). […] While the *α*+ and AI were both higher for free compared to forced trials, median AIs were negative for both free and forced choices (see Appendix 1 for full results; Appendix 1—figure 12).”

Methods:

“FourLR Model*.* In our task, participants made both free and forced choices. Past research suggests that valence biases in learning may differ as a function of choice agency (Cockburn et al. 2014; Chambon et al. 2020). To test this possibility, we assessed the fit of a Four Learning Rate (FourLR) model, which was the same as the RSTD model except that it included four learning rates instead of two, with separate *α*^+^ and *α*^-^ parameters for free and forced choices.”

The preceding explanation is included in Appendix 1 (Appendix 1—figure 12).

As the reviewers pointed out, prior research has also found better subsequent memory for memoranda associated with free versus forced choices (Murty et al. 2015; Katzman and Hartley 2020). The present study was not designed to test the effects of agency on memory. The purpose of our forced-choice trials was to ensure all participants had similar opportunities to learn probabilities and outcomes associated with each probabilistic and deterministic point machine, regardless of their risk preferences. However, to test whether memory was different for items that appeared with outcomes of free versus forced choices, we ran two additional glmer models. First, we ran the most complex model included in the original manuscript (the model predicting memory accuracy that tested for a PE valence x PE magnitude x AI interaction, and also included linear and quadratic age and memory trial number), adding a predictor that indicating whether the image appeared after a free or forced choice. Memory did not vary as a function of choice agency (*z* = -0.43, *p* = .667, *OR* = 0.99, 95% CI: [0.95-1.04]).

As a final test for a potential agency benefit on memory, we tested whether the 3-way PE Valence x PE Magnitude x AI interaction predicting memory performance varied as a function of whether the choice was forced or free (i.e., we tested for a 4-way PE Valence x PE Magnitude x AI x choice agency interaction). The 4-way interaction was also not significant (*z* = -1.39, *p* = .165, *OR* = 0.96, 95% CI: [0.91, 1.02]). To further explore whether agency had any apparent qualitative effect, we plotted the 3-way interaction effect — PE Valence x PE Magnitude x AI) separately for free versus forced choice trials (Appendix 1—figure 13). Strikingly, we found that for free-choice trials (Appendix 1—figure 13A), which comprised the majority of trials, the interaction looks qualitatively similar to the overall 3-way interaction effect across all trials that we report in the main text *(*Figure 4B in the manuscript, shown in Appendix 1—figure 13C). However, for forced trials (Appendix 1—figure 13B), memory performance for images that coincided with positive Pes did not appear to be modulated by AI. This pattern is similar to the memory effect observed in Experiment 2 (Figure 6B in the manuscript, shown in Appendix 1—figure 13D, in which participants provided explicit predictions of outcomes, but did not make choices. This qualitative similarity suggests that memory for surprising high-magnitude positive outcomes may be equivalently facilitated for all individuals in situations where they do *not* have agency, regardless of their idiosyncratic biases in learning.

We added text to the Results and Discussion sections describing prior findings on agency and memory in relation to our study. We also added the multilevel models described above to Appendix 1, with the free vs. forced choice predictor as a main effect as well as the model with the 4-way PE Valence x PE Magnitude x AI x choice agency interaction:

Results:

“Finally, we tested for effects of agency – whether an image coincided with the outcome of a free or forced choice – on memory performance. We did not find a significant main effect of agency on memory, and agency did not significantly modulate the AI x PE magnitude x PE valence interaction effect (see Appendix 1 for full results).”

We have now added a paragraph to the discussion giving a fuller account of the idea that learning and memory may differ as a function of whether choices are self-determined or imposed. In this paragraph, we also discuss observed parallels between the patterns of effects observed on forced choice trials, and those from Experiment 2 in which participants were not able to make choices, and speculate that these patterns may reflect common effects of lack of agency on learning and memory. The corresponding revised sections are excerpted below:

Discussion:

“In Experiment 1 of the present study, participants observed the outcomes of both free and forced choices. […] Thus, while our study was not explicitly designed to test for such effects, this preliminary evidence suggests that choice agency may modulate the relation between valence biases in learning and corresponding biases in long-term memory, a hypothesis that should be directly assessed in future studies.”

3. Please include additional data visualizations that depict the raw data, which will better equip the reader for interpreting the results (e.g., the learning curves per experimental condition, and/or per age group as suggested by Reviewer 2).

We added several new figures that depict the raw data, which we hope will aid in interpreting our results.

First, we now include a depiction of mean accuracy on test trials (where one option dominated the other, and thus there was a “correct” choice) across the task, separately by age group (Appendix 1—figure 1A). This plot clearly reveals that there were no age differences in learning trajectory or asymptotic accuracy.

Reviewer 2 suggested plotting learning curves by experimental condition. We want to clarify that our experiment does not involve different conditions. Rather, participants encounter one of a number of pairs of possible options. In some pairs, as in those plotted Appendix 1—figure 1A, there is a correct option, allowing us to index accuracy and assess learning. However, many of our pairs did not involve a “correct” choice, so there is no way to index accuracy on these trials. That is, in risk trials, one option was risky and one was safe, but neither option strictly dominated the other. For these trials, we plotted mean risk taking with age and block separately for equal EV trials (20 vs. 0/40 or 40 vs. 0/80; Appendix 1—figure 1B) and unequal EV trials (20 vs. 0/80; Appendix 1—figure 1C).

In addition to plotting learning curves, we added several plots that illustrate patterns of memory performance beyond the logistic regression analysis that is the primary focus within the manuscript. We include Appendix 1—figure 3, which depicts the significant relationship between false alarms and age, and the marginally significant relationship between d’ and age, patterns that are reported in the manuscript.

4. Please consider the role of attention versus reward prediction error (RPE) in the memory effects. Reviewers pointed out that there may be attentional differences across probabilistic versus deterministic trials that could be driving the effects rather than RPE, which if the source of memory differences would warrant a different conclusion than the current paper. In addition, whether and how attention fits into the relationship between asymmetry index and memory was unclear. The authors may be in a position to address this question by performing additional analyses on the current data (e.g., by assessing whether RPE strength alone is related to encoding within the probabilistic trials, as suggested by Reviewer 2); or more generally (in the absence of data) clarify their position or speculate on these issues.

As our study did not include attentional measures, we did not focus on the role of attention in the discussion of our results in the initial manuscript. However, given a large literature demonstrating the critical role of attention in reinforcement learning (Pearce and Hall 1980; Dayan et al. 2000; Holland and Schiffino 2016; Radulescu et al. 2019) and memory formation (Chun and Turk-Browne 2007), we agree completely with the reviewers’ intuitions that attention likely played a critical role in the effects we observed.

Prominent theoretical accounts have proposed that attention should be preferentially allocated to stimuli that are more uncertain (Pearce and Hall 1980; Dayan et al. 2000). Computational formalizations of such attentional modulation of learning posit that greater attention might be reflected by increases in learning rates. Within these models, attention to a stimulus is proportional to the aggregate prediction errors it is associated with. Thus, the notion that greater attention might be paid to the outcomes of probabilistic versus deterministic machines in our task has strong theoretical support. However, rather than prediction errors and attention representing distinct potential mechanisms that might influence learning and memory, these models conceptualize prediction errors as direct drivers of attention allocation.

In our study, memory was better for items that coincided with probabilistic compared to deterministic outcomes. We think this finding quite likely reflects greater attention to the episodic features associated with outcomes of uncertain choices. Importantly however, our memory findings cannot be solely explained via an uncertainty-driven attention account. As suggested by reviewers, we tested whether differences in memory for outcomes of deterministic versus probabilistic trials were driving the AI x PE magnitude x PE valence effect observed in our Experiment 1 by re-running the regression only within the subset of trials in which participants made probabilistic choices. Our results did not change — we observed both a main effect of PE magnitude (*z* = 2.22, *p* = .026, *OR* = 1.11, 95% CI [1.01,1.23], *N* = 62) and a significant PE valence x PE magnitude x AI interaction (*z* = 2.34, *p* = .019, *OR* = 1.11, 95% CI [1.02,1.21], *N* = 62). While the absence of attentional measures in the current study precludes decisive inferences about potential attentional mechanisms, this analysis suggests that our current results may reflect differential attention to valenced outcomes that varies systematically across individuals in a manner that can be accounted for by asymmetries in their learning rates. Such valence biases in attention have been widely observed in clinical populations (Mogg and Bradley 2016; Bar-Haim et al. 2007), and may also be individually variable within non-clinical populations.

We have now revised our manuscript to include greater discussion of this putative role of attention in our learning and memory findings:

Introduction:

“Outcomes that violate our expectations might also be particularly valuable to remember. […] Moreover, while few studies have explored the development of these interactive learning systems, a recent empirical study observing an effect of prediction errors on recognition memory in adolescents, but not adults (Davidow et al., 2016), suggests that the influence of reinforcement learning signals on memory may be differentially tuned across development.”

Results:

“We next tested whether the decision context in which images were presented influenced memory encoding. […] This result suggests that pictures were better remembered when they followed the choice of a machine that consistently generated reward prediction errors, which may reflect preferential allocation of attention toward outcomes of uncertain choices (Pearce and Hall 1980; Dayan et al. 2000).”

Results:

“To test whether differences in memory for outcomes of deterministic versus probabilistic trials might have driven the observed AI x PE magnitude x PE valence interaction effect, we re-ran the regression model only within the subset of trials in which participants made probabilistic choices. Our results did not change — we observed both a main effect of PE magnitude (*z* = 2.22, *p* = .026, *OR* = 1.11, 95% CI [1.01,1.23], *N* = 62) and a significant PE valence x PE magnitude x AI interaction (*z* = 2.34, *p* = .019, *OR* = 1.11, 95% CI [1.02,1.21], *N* = 62).”

Discussion:

“Attention likely played a critical role in the observed learning and memory effects. […] Such valence biases in attention have been widely observed in clinical disorders (Mogg and Bradley 2016; Bar-Haim et al. 2007), and may also be individually variable within non-clinical populations.”

5. Please improve the integration of Experiment 2 into the paper. More details were needed for understanding the purpose of this sample. In addition, Reviewers noted that the interaction found in Experiment 1 did not entirely replicate in Experiment 2. This warrants more consideration in the paper.

In our revised manuscript, we have attempted to better integrate Experiment 2 into the manuscript. Specifically, we added additional text clarifying that our motivation was to replicate the effect of valence biases in learning on subsequent memory in an independent and larger sample. Additionally, we wondered whether the observed effect was sufficiently robust that it would be evident when participants were not explicitly making choices (i.e., in a task in which participants made predictions about expected outcomes, but could not choose them) and when learning biases reflected idiosyncratic individual differences across a sample of adults, rather than age-related variation.

We also revised our results and Discussion sections to directly acknowledge the qualitative differences in the interaction patterns observed in Experiments 1 and 2. Specifically, we observed a qualitative similarity between the interaction in Experiment 2, in which participants made explicit predictions but not free choices, and the analysis of the forced-choice trials only in Experiment 1 (see response to Essential Revision 2 above for more details). In both analyses, differences in memory performance as a function of AI were primarily apparent for images coinciding with negative PEs — those who learned more from negative PEs also had better episodic memory for images that coincided with increasingly large negative PEs, while all participants appeared to have stronger memory for images coinciding with larger positive PEs, independent of AI. Based on this qualitative correspondence between the patterns of results, we now introduce a speculative interpretation that the effects of PE on memory may differ in agentic and non-agentic learning contexts.

To better integrate Experiment 2 into the manuscript, we amended the manuscript as follows:

Introduction:

“To determine whether this hypothesized correspondence between valence biases in learning and memory generalized across experimental tasks and samples of different ages, in Experiment 2, we conducted a re-analysis of data from a previous study (Rouhani et al., 2018). […] Here, we tested whether a valence-dependent effect of PE on memory might be evident after accounting for idiosyncratic valence biases in learning.”

Results:

“Next, we assessed the generalizability of the observed effect of valence biases in learning on memory by conducting a reanalysis of a previously published independent dataset from a study that used a different experimental task in an adult sample (Rouhani et al., 2018). […] Here, we examined whether signed valence-specific effects might be evident when we account for individual differences in valence biases in learning.”

Results:

“Consistent with the results reported in the original manuscript (Rouhani et al., 2018), as well as the findings in Experiment 1, there was a strong main effect of unsigned PE (i.e., PE magnitude) on memory (*z* = 5.09, *p* <.001, *OR* = 1.19, 95% CI [1.12, 1.28]). […] One possibility is that PE magnitude and PE valence enhance memory through separate mechanisms, with a universal positive effect of unsigned PEs but a contextually (depending on choice agency) and individually variable effect of PE valence.”

Discussion:

“Studies have also found that subsequent memory is facilitated for images associated with free, relative to forced, choices (Murty et al. 2015; Katzman and Hartley 2020). […] Thus, while our study was not explicitly designed to test for such effects, this preliminary evidence suggests that choice agency may modulate the relation between valence biases in learning and corresponding biases in long-term memory, a hypothesis that should be directly assessed in future studies.”

6. Please clarify the task and methods descriptions. For example, it is not clear whether the forced choices were included in the modelling. Moreover, please ensure all details needed to understand the major points of the paper are included in the main text without requiring readers to reference the methods.

We now include additional text to clarify important methodological details within the Results section. We also modified the text in the Methods section to ensure that our descriptions of the methods were clear and complete. As part of this revision, we clarified that the outcomes of the forced trials were included in the value updating step within the model, but these trials were not included in the modeling stage in which learned values are passed through a softmax to determine choice probabilities, as there was only a single choice option on these trials.

Results:

“Participants (*N* = 62) ages 8-27 *(M* = 17.63, *SD* = 5.76) completed a risk-sensitive reinforcement learning (RL) task (Niv et al., 2012). […] A subsequent memory test allowed us to explore the interaction between choice outcomes and memory encoding across age (Figure 1C).”

Results:

“Next, we explored whether valence biases in learning could account for individual variability in subsequent memory. […] We also tested for effects of linear and quadratic age, false alarm rate, as a measure of participants’ tendency to generally deem items as old, and trial number in the memory task, to account for fatigue as the task progressed (Figure 4A). ”

Methods:

“In all models, Q-values were converted to probabilities of choosing each option in a trial using the softmax rule, P_M1_ = e^β*Q(t)M1^/(e^β*Q(t)M1^+ e^β*Q(t)M2^), where P_M1_ is the predicted probability of choosing Machine 1, with the inverse temperature parameter *β* capturing how sensitive an individual’s choices are to the difference in value between the two machines. Notably, outcomes of the forced trials were included in the value updating step for each model. However, forced trials were not included in the modeling stage in which learned values are passed through the softmax function to determine choice probabilities, as there was only a single choice option on these trials.”

7. As highlighted by Reviewer 3, it is unclear whether asymmetry biases are a trait-stable characteristic, or rather would change with age within an individual. Please speak to this point as well as other limitations associated with the cross-sectional nature of this study in the revised Discussion.

We agree with the editor and reviewer’s suggestion that the manuscript would benefit from a fuller discussion of how the age differences in valence asymmetry biases that we observed in our cross-sectional sample might be interpreted as reflecting stable or malleable individual differences, as well as the limitations of the inferences that can be made from cross-sectional results.

Past studies have demonstrated that valence asymmetries in reinforcement learning are malleable within a given individual, exhibiting sensitivity to the statistics of the environment (e.g., the informativeness of positive versus negative outcomes (Pulcu and Browning 2017)) as well as endogenous manipulations such as the pharmacological manipulation of neuromodulatory systems (Michely et al. 2020). In Experiment 1, we aimed to quantify age-related variation in valence biases in learning and memory in a task in which there was no specific level of bias that yielded optimal performance. One of our central motivations for adopting this design was the theoretical suggestion that individual valence biases in learning should be malleable across contexts—that they should adapt to the reward statistics of the environment in a manner that maximizes reward (Cazé and Meer 2013). While prior developmental studies have reported age differences in valence biases across adolescence (Christakou et al. 2013; Van Den Bos et al. 2012; Hauser et al. 2015; Jones et al. 2014; Master et al. 2020), the interpretation of these findings confounded. They may reflect age-differences in an adaptive reward maximization process, rather than performance-independent, age-associated biases (Nussenbaum and Hartley 2019).

Here, we found that independent of this optimality confound, valence asymmetries in our cross-sectional sample varied nonlinearly as a function of age. These findings raise the possible suggestion that, for a given individual, learning rate asymmetries might become more negative from childhood into adolescence, and more positive from adolescence into young adulthood. However, such predicted patterns of change cannot be validly inferred from cross-sectional studies, which are confounded by potential effects of cohort (Schaie 1965). Future longitudinal studies would be needed to definitively establish whether such age-related changes can be observed within an individual over developmental time.

We have made the following corresponding modifications to the manuscript:

Discussion:

“The present findings raise the suggestion that, for a given individual, valence asymmetries in value-based learning might become more negative from childhood into adolescence, and more positive from adolescence into young adulthood. […] Future longitudinal studies will be needed to definitively establish whether valence biases in learning exhibit systematic age-related changes within an individual over developmental time.”

8. The Reviewers had several additional thoughtful suggestions for other work the authors might cite. I recommend the authors consider whether these suggestions might enhance the novelty, impact, and/or clarity of their paper.

We thank the reviewers for their thoughtful suggestions on additional citations. We added these citations as follows:

In response to Reviewer 2, we noted that Master et al. (2019) examined reinforcement learning across adolescence, and demonstrated increases in negative learning from childhood into adolescence:

Introduction:

“Several past studies have characterized developmental changes in learning from valenced outcomes (Christakou et al. 2013; Hauser et al. 2015; Jones et al. 2014; Master et al. 2020; Moutoussis et al. 2018; Van Den Bos et al. 2012).”

Discussion:

“Moreover, heightened reactivity to negatively valenced stimuli has also been observed in adolescents, relative to children (Master et al. 2020) and adults (Galván & McGlennen, 2013). While a relatively small number of studies have used reinforcement learning models to characterize age-related changes in valence-specific value updating (Christakou et al. 2013; Hauser et al. 2015; Jones et al. 2014; Master et al. 2020; Moutoussis et al. 2018; Van Den Bos et al. 2012), age patterns reported in these studies vary substantially and none observed the same pattern of valence asymmetries present in our data.”

As suggested by Reviewer 3, we briefly described the paradigm in Van den Bos et al. (2012) to demonstrate the novel contribution of the current study:

“Several past studies have characterized developmental changes in learning from valenced outcomes (Christakou et al. 2013; Hauser et al. 2015; Jones et al. 2014; Master et al. 2020; Moutoussis et al. 2018; Van Den Bos et al. 2012). […] Thus, choice behavior in these studies might reflect both potential age differences in the optimality of reinforcement learning, as well as context-independent differences in the weighting of positive versus negative prediction errors (Cazé & van der Meer, 2013; Nussenbaum & Hartley, 2019).”

As described in response to Essential Revision 2, we now include analyses of memory as a function of free compared to forced choices, inspired in part by our prior work, along with relevant citations (Katzman & Hartley, 2020; Murty et al., 2015):

Discussion:

“Prior studies have demonstrated differential effects of free versus forced choices on both learning and memory (Katzman and Hartley 2020; Cockburn et al. 2014; Chambon et al. 2020), which may reflect greater allocation of attention to contexts in which individuals have agency. […] Studies have also found that subsequent memory is facilitated for images associated with free, relative to forced, choices (Katzman and Hartley 2020; Murty et al. 2015).”

9. Please provide additional details on the models (e.g., reporting parameter β, age effects, model validation, and/or more information on the recoverability analysis) as appropriate. One specific question raised by reviewers was whether learning rate parameter estimation might be more difficult in participants with more deterministic choices and thus fewer trials with non-zero RPEs. Is this empirically true, i.e., does model parameter recovery (as shown in Table 1 overall) differ across ages/ranges of behavior?

Our revised manuscript and supplement now include additional information about differences in parameter estimates and recoverability across age and behavioral patterns. First, as suggested by the reviewers, we assessed whether *β* differed with age in the RSTD model. We found that *β* increased with age numerically, but there was no statistically significant relation with age (Appendix 1—figure 10C; *r* = .18, *p* = .156).

Next, we tested whether parameter recovery differed as a function of individual differences in the propensity to make deterministic choices. Specifically, we tested whether parameter recovery and model fit (BIC) varied as a function of AI. Importantly, because AI is so highly correlated with risk taking (as discussed in response to Reviewer 1, Essential Revision 1), this analysis allows us to address the reviewers’ questions about potential differences in parameter recoverability for participants who more frequently chose the deterministic point machines (i.e., those with low AI). To this end, we divided the simulated participants into AI quartiles and examined parameter recovery and model fit in each AI quartile. We found that the parameters were reasonably well recovered at all levels of AI (Appendix 1—figure 8A, B, and C.).

However, parameter recoverability varied across levels of AI, in a manner that runs somewhat counter to the reviewers’ intuitions. In particular, recovery of the *α*^+^ parameter was relatively poorer for the simulated participants in the High AI quartile (Appendix 1—figure 8A) and *α*^-^ recoverability was relatively poorer for those in the Low-AI quartile (Appendix 1—figure 8A). Taken together, these patterns suggest that learning rate parameters are relatively less well recovered for individuals with higher AIs (i.e., who made more risk-seeking choices).

This differential recoverability as a function of AI stems from the interactions between subjects’ risk preferences and the set of risky choice trials presented in our task. There were two types of risky trials in our task: Equal-EV, (0/40 vs. 20, or 0/80 vs. 40) and Unequal-EV (0/80 vs. 20). This particular combination of Equal- and Unequal-EV risk trials led to differential resolution in the estimation of valenced learning rates as a function of AI. Learning rates for risk-averse participants could be estimated more accurately because those who were very risk averse (and thus had a much larger *α*^-^ than *α*^+^) might choose both the safe 40 point option and the safe 20 point option over the 0/80 machine, whereas those who were less risk averse might prefer the safe 40 to the 0/80, but the 0/80 over the safe 20. In contrast, those with high positive AI are likely to choose the risky option on every Equal and Unequal-EV trial, so the ability to distinguish precisely between different levels of *α*^+^ for those with high AI is diminished. Our model fit results provide further evidence that this lower *α*^+^ recoverability in individuals with high AIs stems from this aspect of our task structure (Appendix 1—figure 8D). Despite the model’s imprecise *α*^+^ estimation for these high-AI subjects, the model was able to predict behavior well (i.e., BIC was low), likely because these participants are likely to take risks on all Equal and Unequal-EV risk trials, regardless of their precise *α*^+^ level.

This increase in parameter recovery for those participants with low compared to high AI runs counter to the reviewers’ intuitions that smaller PEs may drive worse recovery in these Low-AI participants. Although it is true that PEs were smaller for those with relatively lower AIs (see Appendix 1—figure 11F in response to Essential Revision 1), our design required these low-AI participants to experience high-magnitude PEs from risky choices on some trials. Specifically, in our task, participants encountered forced trials, where participants were required to choose specific machines, which were sometimes risky, and Test trials, where one option dominated the other, and the dominating option was sometimes risky. Thus, including these Forced and Test trials may have boosted our ability to recover learning rates in those with Low AI, while also providing opportunities for participants to sufficiently learn and demonstrate their knowledge of machine outcomes and probabilities.

Despite these differences in parameter recoverability at different levels of AI within this experimental paradigm, there are several reasons why we don’t believe that these results are problematic for interpreting the current results. First, these simulations were generated by sampling uniformly from the range of learning rate and temperature parameters observed in our empirical sample. Thus, these simulated participants can take on AI levels that are not actually represented within our empirical sample. Indeed, 81% of fit AI values observed in our empirical sample fall within the lower three quartiles of AI values for this simulation, for which parameter recoverability was higher. Moreover, the recoverability estimates in this simulation dramatically overrepresent participants with low levels of decision noise relative to our empirical sample, which distorts these estimates of recoverability to be lower than what would actually be obtained for our empirical sample of participants (e.g., when we exclude any simulated participants with decision noise < 2, which captures the vast majority of participants in our sample, recoverability of *r* values all increase by ~.05).

To address this comment in the manuscript, we now report the analysis of age differences in the *β* parameter to the Results section and have added these new recoverability analyses to Appendix 1, and we reference these analyses within the Methods section.

Results:

“We computed an asymmetry index (AI) for each participant, which reflects the relative size of *α*^+^ and *α^-^*, from the RSTD model. […] Finally, there were no significant linear or quadratic age patterns in the *β* parameter (*p*s >.15, see Appendix 1 for full results; Appendix 1—figure 10C).”

Methods:

“We also found that RSTD model parameters were reasonably well recovered across the range of AI observed in our empirical sample (see Appendix 1, Appendix 1—figure 8).”

Appendix 1:

“As reported in the main text, the relation between AI and age appears to be driven by quadratic age patterns in *α*^-^ (*b* = -.09, 95% CI [-0.16, -0.03], *t*(59) = -3.01, *p* = .004, *f*^2^ = 0.15, 95% CI [0.02, 0.43]; Appendix 1—figure 10B). […] Additionally, the relation between age *β* was not significant (*b* = .56, 95% CI [-0.22, 1.33], *t*(60) = 1.44, *p* = .156, *f*^2^ = 0.03, 95% CI [0, 0.19]; Appendix 1—figure 10C).”

Finally, we included the detailed description of recovery as a function of AI in Appendix 1 (p. 33-34, lines 1259-1325, Appendix 1—figure 8).

Reviewer #1:[…] 1. I am left unsure as to the value added by the reinforcement learning model and whether the asymmetry index reflects something unique about learning as compared with a general bias towards deterministic choices in the adolescent period. That is, if the authors wish to make claims about asymmetry of learning rate specifically (α) it would be important to know if this is dissociable from the choice or sampling bias they observe. Empirically, is it the case that there would be a similar set of findings if one focused on the percent probabilistic choices value per participant (data in Figure 2) rather than the asymmetry index (Figure 3)? If these are somewhat redundant or overlapping metrics, I would encourage the authors to clarify this in the paper and perhaps only focus on one of them in the main text, so as not to imply these are separate findings.

We thank the reviewer for this insightful comment. It is true that asymmetry index is highly correlated with percent probabilistic choices. However, we feel that the RL models are valuable in that they enable us to test hypotheses about putative decision processes that drive risk taking. This point is perhaps even clearer in the revised manuscript, in which we now include two additional models of the learning process within our model comparison. Moreover, although AI and percent risky choice were highly correlated, the models provided additional value in enabling us to test the hypothesis that learning biases might be related to subsequent memory. The formal RL models are necessary for computing the individually varying trial-by-trial PEs for each participant that we use as predictors for the memory analysis.

To further demonstrate the explanatory value of PEs derived from RL models, we ran an analysis analogous to the multilevel model described throughout our paper, but without measures derived from RL models. We tested for a 3-way interaction in subsequent memory between a participant’s percent risk taking, the outcome magnitude (similar to PE magnitude, coded as the absolute value of the outcome magnitude minus 40), and whether the outcome value was greater than or equal to 40 versus less than 40 (similar to PE valence). Although there was a significant effect of outcome magnitude, such that images coinciding with more extreme outcomes (0 and 80 points) were better remembered than medium point outcomes, similar to the PE magnitude effect in memory, the 3-way interaction was not significant (*p* = .638). We included a plot of the 3-way interaction in Author response image 2, which shows a different pattern from the 3-way interaction effect observed using PEs and AIs derived from the RSTD model. Collectively, our findings suggest that the RL models are helpful in understanding the mechanisms underlying choice, and yield measures (PEs and AIs) that can better explain individually varying valence biases in both learning and memory.

**Author response image 2. sa2fig2:** Non-significant results from a 3-way interaction in subsequent memory data using individual difference and outcome value measures not derived from a reinforcement learning model.

2. Related to my above point, those individuals who make fewer probabilistic choices (disproportionately adolescents) have fewer opportunities to learn from prediction errors that are either positive or negative. That is, in my understanding, there will be no prediction error from the deterministic machines which the adolescents primarily select. Their bias to select deterministic machines seems fairly extreme; it appears as though they shy away from the probabilistic choices across the board, even when the value of the probabilistic choice on average would greatly exceed that of the deterministic choice (e.g., as shown in Figure S2). As such, I am wondering whether theoretically the authors can clarify the relationship between choice and learning rate; and analytically whether the bias towards deterministic choices could impact the model fitting procedures.

We appreciate this valuable comment. We addressed this comment in response to Essential Revision 9. Briefly, it is true that those with more negative learning biases also make fewer probabilistic choices on “risk” trials – that is, trials in which they face a choice between a risky and a safe option, neither of which dominates the other. However, these participants have opportunities to experience large PEs on trials where probabilistic machines are the only option (i.e. “forced” choices) and on some “test” trials where a probabilistic machine is the dominating option.

We analytically tested whether biases toward deterministic choices might have impeded model fitting for risk-averse participants. Counterintuitively, we found somewhat *better* parameter recovery for those biased toward safe choices. Please see Essential Revision 9 for a detailed summary of this analysis.

3. As an additional interpretive question, I had trouble linking the asymmetry index to the memory performance and understanding how the authors believed these metrics to be related. Is there a general increase in interest or attention or salience when a prediction error aligning with the learner's biases (e.g., a highly positive prediction error if I'm someone who learns best from those types of trials – that is, showing a positive AI) happens, therefore indirectly impacting memory formation in the process? Or, is this thought to be a separate mechanism that occurs (the learning from positive prediction errors vs. "prioritization" in memory of those positively valenced experiences)? It seems to me as though both are related to learning/encoding, and thus this individual difference may not be terribly surprising, but I was unsure about the authors' preferred interpretation.

We appreciate this insightful comment. We agree that individual differences in value computation and memory performance may likely reflect individually varying valence biases in attentional allocation. Such valence biases in attention have been widely observed in clinical disorders (Mogg and Bradley 2016; Bar-Haim et al. 2007), and may also be evident within non-clinical populations.

Please see our response to Essential Revision 4, where we now describe in greater detail hypotheses regarding the role of attention in our findings, and our manuscript revisions that discuss such potential attentional mechanisms.

4. While I appreciated the inclusion of experiment 2, I felt that it was not particularly well integrated into the paper. For example, why did the authors opt to use data with only adults rather than a developmental sample (and does this constrain interpretation in any way)? In addition, it is important to highlight that the results do not fully replicate the main findings. In particular, there is no difference in the relationship between the magnitude of prediction error and memory among positively valenced trials according to AI, which was observed in the main developmental experiment 1. This discrepancy warrants more attention in the paper. (Could this be about the sample characteristics? Task differences? and so on.)

We thank the reviewer for this helpful comment. In brief, our high-level motivation for Experiment 2 was to replicate the effect of valence biases in learning on subsequent memory in an independent and larger sample. While the use of this particular dataset was admittedly opportunistic (this was the only dataset amenable to RL with trial-unique memoranda that we were able to gain access to), we were interested in testing whether the observed effect was sufficiently robust that it would be evident when participants were not explicitly making choices (i.e., in a task in which participants made predictions about expected outcomes but could not choose them) and when learning biases reflected idiosyncratic individual differences across a sample of adults, rather than age-related variation. Please see our response to Essential Revision 5, where we describe our improved integration of Experiment 2 into the manuscript.

Additionally, we now more explicitly acknowledge that Experiment 2 does not fully replicate Experiment 1. In addressing Essential Revision 2 about agency and the observed individual differences in memory, we discovered that the memory patterns in forced trials of Experiment 1 (i.e., trials where participants were only given one option and therefore did not make an autonomous choice) are similar to the pattern observed in Experiment 2, where participants made predictions rather than choices. We therefore hypothesize that the discrepancy between memory patterns in Experiments 1 and 2 stem from differences in task demands, specifically with respect to making agentic choices. Please see our response to Essential Revision 2, where we speculate on how agentic and non-agentic choices may yield different memory patterns, a point we now raise in the discussion.

5. It was not clear at the conceptual level how the Pavlovian reinforcement learning model fit in experiment 2 is different from the main RSTD used in experiment 1, and/or why these paradigms required the use of different models. Additional description on this point would be helpful.

In Essential Revision 1, we detailed the differences between the task in Experiment 1, where participants made explicit choices on most trials, and that of Experiment 2, where participants made explicit outcome value predictions on each trial. Because the behavioral measures differed across the 2 experiments (one model was fit to choices while the other was fit to value estimates), we needed to specify slightly different models for the tasks.

6. I would appreciate more context for the recoverability results. In particular, the ability to recover which model generated the data for simulated participants (RSTD vs. TD) seemed fairly low. I understand the point about low-asymmetry participants generated by the RSTD model being more parsimoniously fit by the simpler TD model, so that direction of error does not bother me so much. However, I am puzzled by the 87% correct for those simulated participants generated by the TD model. This would have to mean, I think, that the simple TD behavior was being incorrectly attributed to the more complex two-alpha model? I was hoping the authors could provide additional context or reassurance that this qualifies as "good" performance.

We thank the reviewer for raising this concern. As the reviewer notes, while it is clear why a proportion of the simulated participants generated with the RSTD model were better fit by the TD model (i.e., those with low asymmetry), it was indeed somewhat puzzling why only 87% of those participants simulated with the more parsimonious TD model were better fit by the more complex two-alpha model. To better understand why this was the case, we closely examined the simulated and estimated parameters to see whether there was some systematic bias present in a subset of the TD-simulations that enabled the single learning-rate model to effectively reproduce choice behavior that would more commonly be generated by the RSTD model. We found no such systematic effects. Instead, these participants exhibited some small degree of risk-seeing or risk-averse choice bias and correspondingly, had a small AIC difference between the two models that, while essentially equivocal, favored the RSTD model. We took this to be potential evidence of overfitting. We reasoned that if the choice behavior generated by models in this task were indeed differentiable but the AIC metric simply did not apply a sufficient penalty for the additional free parameter, that adopting a more conservative metric (i.e., one that requires a greater improvement in fit in order to justify the addition of a parameter) would correct this poor model recoverability performance. Indeed, when we used the Bayesian Information Criterion metric instead, we found that the proportion of TD-simulated participants best fit by the TD model was now 98%. To correct for potential overfitting in our results, we have now moved to use this more conservative BIC metric in all of our model comparisons. Importantly, none of the conclusions of our model comparisons are affected by this change. We are grateful to the reviewer for pushing us to better understand and correct this issue.

7. There were some details not sufficiently described in the main paper for the reader to understand without referencing the methods or supplement. For example: How did the forced versus choice trials work? What is "test trial performance" – the test trials are not described until the supplement and it was confusing to me because the only "test" mentioned in the main paper at this point was the memory test. How were the probabilistic vs. deterministic choices incorporated into the mixed-effects modeling?

We thank the reviewer for raising this point. Please see our response to Essential Revision 6, where we describe revisions to the Results section. Specifically, we modified our description of the choice task within the main text to include the trial types (risk trials, test trials, forced trials) that participants encountered. We also modified our description of the mixed-effects modeling to clarify that whether a choice was probabilistic or deterministic was not included in the model. Rather, probabilistic and deterministic choices are reflected in the PE magnitude predictor variable, where, in general, more extreme PE magnitudes reflect outcomes of risky choices, and over time, deterministic choices do not elicit PEs at all.

8. How were the different responses on the memory test considered? There were four choices according to the figure but it is described as though it is simply "old" vs. "new" responses. Please clarify how this was handled in analysis as well as the reason for inclusion of these four options.

We appreciate this helpful comment. Our focus in the present manuscript was on memory accuracy rather than confidence, so we collapsed across confidence ratings (e.g., “Definitely old” and “Maybe old”), a convention widely adopted in manuscripts examining memory accuracy effects (e.g., Dunsmoor et al. 2015; Murty et al. 2016).

However, based on the reviewer’s suggestion, we now include an ordinal regression analysis in Appendix 1 that tests for an AI x PE Valence x PE Magnitude interaction but with participants’ uncollapsed responses (1 = Definitely New, 2 = Maybe New, 3 = Maybe Old, and 4 = Definitely Old).

Regression results are reported in Appendix 1—table 1. Importantly, the results from the ordinal regression were not meaningfully different from results collapsed across confidence ratings. In particular, the AI x PE Valence x PE Magnitude interaction was highly significant in the ordinal regression. The 3-way interaction is plotted in Appendix 1—figure 4*.* Here, probabilities of each memory response are plotted as a function of PE valence and magnitude separately for AI = -.8 (top panels), AI = 0 (middle panels) and AI = .8 (bottom panels). Consistent with the results from the model that collapsed across confidence, the likelihood of a “definitely old” response was highest for those in those with low AI for images that coincided with high-magnitude negative PEs (top left panel) and those with high AIs for images that coincided with high-magnitude positive PEs (bottom right panel).

We now include these ordinal regression results in Appendix 1 and briefly mention the analysis in the main text:

Results:

“We had no a priori hypothesis about how any effects of valence bias on memory might interact with participants’ confidence in their “old” and “new” judgments. […] Notably, neither linear (z = .30, p = .767, OR = 1.02, 95% CI [0.89, 1.17]) nor quadratic age (z = -0.15, p = .881, OR = 0.99, 95% CI [0.85, 1.16]) were significant predictors of memory, suggesting that AI parsimoniously accounted for individual differences in memory.

Appendix 1:

“Our multilevel models of memory data collapsed across confidence ratings (e.g., “Definitely old” and “Maybe old”), a convention widely adopted in manuscripts examining memory accuracy effects (e.g., Dunsmoor et al. 2015; Murty et al. 2016). […] Consistent with the results reported in the main text, the likelihood of a “definitely old” response was highest for those in those with low AI for images that coincided with high-magnitude negative PEs (top left panel) and those with high AIs for images that coincided with high-magnitude positive PEs (bottom right panel).”

9. I apologized if I missed this point in the paper: I understand that the models were fit to individual data and most participants were better fit by RSTD than TD. However, the authors also discuss the RSTD model winning overall and all subsequent analyses on the individual differences require the two separate α values. So, I am confused as to whether (a) all participants were ultimately fit with the RSTD so all could be included in the analysis, despite some being better fit by TD or (b) only participants who were best fit by RSTD were included in subsequent analyses (or of course (c) something else)? I think this would be OK because my assumption would be that the participants would simply have two α values (positive and negative) that would be similar if their behavior is better explained by the TD model, but I just wanted to clarify.

All participants’ choice data were fit to TD and RSTD (and now Utility and FourLR) models. Even when patients were best fit by the TD model, the learning rates from the RSTD model were used for subsequent analyses. As the reviewer suggested, participants better fit by the TD than the RSTD model had similar *α*^+^ and *α*^-^ values (i.e., their AI was close to 0). This pattern is demonstrated in Appendix 1—figure 6, which displays AI as a function of the relative BIC for RSTD vs. TD models.

Reviewer #2:[…] First the behavioral results are presented in a very processed manner, limiting the ability to re-interpret the results.Second, the computational modeling is fairly limited and lacking in competing accounts, which also limits the interpretation of the results. As such, it seems that at this point, the results do not fully support the conclusions – in particular, it is not clear that the interpretation in terms of asymmetric learning rates is warranted yet.Comments for the authors:1. Presentation of the results. All figures presented are quite removed from the raw data. It would be helpful to provide more unprocessed results – for example, the learning curves per experimental condition, and/or per age group. This would also provide a more sensitive target for model validation than the ones currently presented in Figure S4. It is much harder for the reader to interpret the results when only very processed data is shown.

We appreciate this helpful comment. As described in Essential Revision 3, we added several additional figures to Appendix 1, including learning curves by age group as suggested by the reviewer.

This is a well-written, easy to read article, that adds a new data point to the complex and contradictory literature on valence, risk, and adolescence, without resolving it. I see a number of issues with the presentation of the results, the modeling and interpretation of the results.2. Modeling. The authors use two very simple RL models to capture the performance (a classic δ rule model, and the same with two learning rates). There are a few relevant aspects to the modeling that are either not considered or not reported, but that are important for interpreting the results.a. Please indicate Q-value initialization and reward rescaling as part of the model description, rather than as part of the model fitting procedure. The initialization has theoretical impacts, so should be part of the model.

As suggested by the reviewer, we now indicate that rewards were rescaled to range from 0 to 1, and Q-values were initialized at.5 (equivalent to 40 points), in our description of the model.

b. Prospect theory indicates that a value of 80 is subjectively worth less that 2* the subjective value of 40. As such, the claim that "expected value" and "risk" are controlled for is debatable: if a true 40 feels like a 50 in comparison to an 80, then the two "same objective expected value stimuli" have different subjective expected values, without a need to invoke risk. The authors should test a competing model where the learning rate is fixed, but the obtained reward is modified according to prospect theory (i.e. subjective reward = 80*((reward/80)^p), where p is a free parameter). This model should also be able to capture the presented processed results (existence of an apparent "risk-aversion"), but should have slightly different temporal dynamics, and would lead to a completely different interpretation of the results.

We thank the reviewer for this valuable comment. As suggested, we ran a new model where subjective reward = 80*((reward/80)^p). We call this the Utility model because Prospect theory does not differ from the Utility model in the context of our task, which only includes positive outcomes. Briefly, we did not find clear differences in quantitative model fit between Utility and RSTD models. However, we did find that posterior predictions of choice data from the RSTD model were more accurate than from the Utility model, and that valence biases and PEs from the RSTD model could qualitatively explain subsequent memory patterns, while the Utility model could not. Please see our response to Essential Revision 1B for a description of our Utility model and related results.

c. Please report parameter β, age effects. Also provide more model validation.

Please see our response to Essential Revision 9 for age effects in the β parameter, along with additional model validation.

d. Have the author investigated the effect of forced vs. free choice trials, both on RL and memory? There is evidence that this leads to differential learning processes, and potentially differential attention, which could impact both the learning and memory findings.

We thank the reviewer for this helpful suggestion. In response to this comment, we investigated how agency in decision making may have influenced both RL and memory by testing whether learning and memory patterns varied for free compared to forced choices. A detailed discussion of these analyses can be found in Essential Revision 2. Briefly, in a model that included separate positive and negative learning rates for free and forced choices (i.e., the FourLR model) positive learning rates were numerically greater in free compared to forced choices, but average learning asymmetries for free choices were not positive. Additionally, individual differences in memory for items presented during forced vs. free choices qualitatively differed (although there was no significant effect of choice agency on memory performance), such that memory patterns for forced choices were similar to those observed in Experiment 2, where participants also did not make choices. We have also added to the discussion further consideration of the potential role of choice agency in this study.

3. Memory findings:a. Can the authors comment on the role of attention? Presumably, the participants pay more attention to the outcome of probabilistic than deterministic choices. Could this be a factor in encoding the image, instead of the actual RPE strength? Is there some analysis in probabilistic trials that could be done to show convincingly that the actual size of the RPE matters? In the current analysis, it seems confounded with condition.

We thank the reviewer for this helpful comment. We agree that attention likely plays a mechanistic role in our observed findings, and now include a discussion of the role of attention in the manuscript. Additionally, we found evidence that PE strength matters within the probabilistic trials — that is, our PE valence x PE magnitude x AI interaction remained significant when we removed images that were presented alongside deterministic outcomes. Please see our response to Essential Revision 4 for more information.

b. While the overall statistical pattern is replicated (Figure 4 and 6A), the realization of the triple interaction looks different in the two experiments (Figure 4 and 6B). In the replication, the asymmetry seems to not matter for positive RPEs, and to have a stronger effect for negative RPEs. For the new study, the patterns seem symmetrical between positive and negative RPEs. Can the authors comment on this?

We now explicitly address the statistical patterns observed in Experiments 1 and 2, and include analyses that aid in interpreting the differences in Appendix 1. Specifically, the pattern observed in Experiment 2, where participants made predictions about stimuli rather than choices, mirrors the pattern seen within forced trials in Experiment 1. Therefore, the difference in these patterns of memory performance may relate to the differential degree of choice agency across both trial types (free vs. forced) and experiments. Please see Essential Revisions 2 and 5 for additional information.

Reviewer #3:[…] This paper presents data from a cross-sectional sample. This raises questions as to whether learning asymmetries are a stable individual characteristic, or whether these biases exhibit within-person changes with age. Nonetheless, the results of this paper provide important advances to our understanding of age-related differences in learning, decision making, and memory that can form the basis of future longitudinal studies.

We thank the reviewer for their helpful comments. We now note the weaknesses related to our cross-sectional sample in the Discussion section, and speculate that the biases shift within individuals across age. Please see our response to Essential Revision 7 for a detailed description of these revisions.

Comments for the authors:This manuscript is exceptionally well-written, and the authors present the methods and findings very clearly. I commend the authors approach to computational modeling and nonlinear age analyses. The present findings provide exciting and novel insights about how adolescents approach risky decision making, which in turn has consequences for memory formation. This is a strong paper, and I believe that the current conclusions warrant publication in eLife. However, I also believe that the inclusion of some additional analyses and clarifications, which I offer below, will further strengthen this manuscript.In the introduction, the authors explain how prior research in developmental samples cannot disentangle learning asymmetries because performance in prior tasks improved if individuals relied upon updating from positive prediction errors. To clarify this point, and to emphasize the novelty of the current study, it would be helpful if the authors provide a more detailed explanation as to how the present design differs from the paradigm used in Van den Bos 2012.

We thank the reviewer for this valuable comment. As suggested, we included a description of the key differences between our paradigm and that employed in Van den Bos et al., (2012). Please see Essential Revision 8 for our revised text.

In the present paper, the authors run model comparison for a basic TD model and a risk-sensitive reinforcement learning model. However, risk sensitivity may be influenced by nonlinear utility functions. It would be informative if the authors also discussed the utility model, as presented in the Niv 2012 paper, which first introduced the present behavioral paradigm. If the authors did not fit this model, please provide an explanation as to why this model was not tested in the current sample.

As suggested by the reviewer, we tested a Utility model as an alternative to the TD and RSTD models. Briefly, we did not find clear differences in quantitative model fit between Utility and RSTD models. However, we did find that posterior predictions of choice data from the RSTD model were more accurate than from the Utility model, and that valence biases and PEs from the RSTD model could qualitatively explain subsequent memory patterns, while the Utility model could not. Please see our response to Essential Revision 1B for a description of our Utility model and related results.

Prior work from this group has shown that choice and agency can influence memory (e.g. Katzman & Hartley 2020). Therefore, for the memory data, it would be helpful if the authors accounted for the different trial types (choice trials vs. forced trials) in the analyses.

We thank the reviewer for this comment. We now include analyses of memory as a function of choice agency. Individual differences in memory for items presented during forced vs. free choices qualitatively differed (although there was no significant effect of choice agency on memory performance), such that memory patterns for forced choices were similar to those observed in Experiment 2, where participants also did not make choices. We have also added to the discussion further consideration of the potential role of choice agency in this study. Please see Essential Revision 2 for additional details.

Due to the cross-sectional nature of the sample, it is unclear if asymmetry biases are a trait-stable characteristic, or whether this bias changes with age within an individual. It woul be helpful for the authors to address this in the discussion.

We agree that our cross-sectional design is a key limitation of our study. We now acknowledge this limitation in the discussion and suggest that future longitudinal studies examine how valence biases in learning and memory shift within an individual with age (please see Essential Revision 7 for more details).